# Real-time tracking reveals catalytic roles for the two DNA binding sites of Rad51

Kentaro Ito [1], Yasuto Murayama [1,5,6], Yumiko Kurokawa [1,5], Shuji Kanamaru [1,2], Yuichi Kokabu [3,7], Takahisa Maki [1], Tsutomu Mikawa [4], Bilge Argunhan [1], Hideo Tsubouchi [1,2], Mitsunori Ikeguchi [3], Masayuki Takahashi [2] & Hiroshi Iwasaki [1,2]✉

During homologous recombination, Rad51 forms a nucleoprotein filament on single-stranded DNA to promote DNA strand exchange. This filament binds to double-stranded DNA (dsDNA), searches for homology, and promotes transfer of the complementary strand, producing a new heteroduplex. Strand exchange proceeds via two distinct three-strand intermediates, C1 and C2. C1 contains the intact donor dsDNA whereas C2 contains newly formed heteroduplex DNA. Here, we show that the conserved DNA binding motifs, loop 1 (L1) and loop 2 (L2) in site I of Rad51, play distinct roles in this process. L1 is involved in formation of the C1 complex whereas L2 mediates the C1–C2 transition, producing the heteroduplex. Another DNA binding motif, site II, serves as the DNA entry position for initial Rad51 filament formation, as well as for donor dsDNA incorporation. Our study provides a comprehensive molecular model for the catalytic process of strand exchange mediated by eukaryotic RecA-family recombinases.

[1] Institute of Innovative Research, Tokyo Institute of Technology, 4259 Nagatsuta, Midori-ku, Yokohama, Kanagawa 226-8503, Japan. [2] School and Graduate School of Bioscience and Biotechnology, Tokyo Institute of Technology, 4259 Nagatsuta, Midori-ku, Yokohama, Kanagawa 226-8503, Japan. [3] Graduate School of Medical Life Science, Yokohama City University, 1-7-29 Suehiro-cho, Tsurumi-ku, Yokohama, Kanagawa 230-0045, Japan. [4] RIKEN Center for Biosystems Dynamics Research, 1-7-22 Suehiro-cho, Tsurumi-ku, Yokohama, Kanagawa 230-0045, Japan. [5] Present address: Center for Frontier Research, National Institute of Genetics, 1111 Yata, Mishima, Shizuoka 411-8540, Japan. [6] Present address: Department of Genetics, SOKENDAI (The Graduate University for Advanced Studies), 1111 Yata, Mishima, Shizuoka 411-8540, Japan. [7] Present address: Department of Bioscience, Mitsui Knowledge Industry, 2-5-1 Atago, Minato-ku, Tokyo 105-6215, Japan. ✉email: hiwasaki@bio.titech.ac.jp

Homologous recombination both generates genetic diversity in meiosis and preserves the integrity of genomic information during the mitotic cell cycle[1,2]. The central step in homologous recombination is the DNA strand exchange reaction between homologous DNA molecules. This reaction is initiated at a single-stranded DNA (ssDNA) region generated through the concerted action of nucleases and helicases at DNA double-strand breaks or at ssDNA gaps generated after passage of the replication fork over damage sites on the template[3–6]. After binding to ssDNA, RecA-family proteins interrogate intact double-stranded DNA (dsDNA) for homology. Once homology is found, the ssDNA invades the dsDNA and displaces the non-complementary strand of the donor to form a new heteroduplex. Strand invasion allows the 3′ end of the broken ssDNA to serve as a primer for the initiation of repair DNA synthesis, thereby recovering lost genetic information by copying it from the donor. Finally, the resultant recombination intermediates are resolved by several enzymatic processes[7–9].

RecA-family recombinases (hereafter referred to as recombinases) are evolutionarily conserved ATPases that catalyze strand exchange during recombination. Recombinases include RecA in bacteria, RadA in archaea, and Rad51 and Dmc1 in eukaryotes[10–13]. Recombinase protomers assemble on ssDNA in an ATP-dependent manner to form a right-handed helical nucleoprotein filament known as the presynaptic filament. This higher-order structure comprises the catalytic core involved in the homology search and subsequent DNA strand exchange reaction[10,14]. Once the non-complementary strand of the donor is released, the recombinase filament wraps around the newly generated heteroduplex DNA, which is often referred to as the postsynaptic filament.

The most extensively studied recombinase is RecA of *Escherichia coli*. In the presynaptic filament, RecA binds three nucleotides per monomer and the ssDNA is stretched to a length 1.5x that of B-form dsDNA. Curiously, the stretching is not uniform, but instead occurs between adjacent RecA monomers, leading to the formation of inter-triplet gaps. Triplets bound to RecA always retain a B-form-like conformation[15].

Each RecA monomer is suggested to have two distinct DNA binding sites, site I and site II, that participate in the strand exchange reaction[14,16]. Site I is considered to bind ssDNA to form the presynaptic filament whereas site II is thought to mediate capture of donor dsDNA and the homology search. Thus, a feasible scenario is that once the presynaptic filament senses homology via site II, a complementary strand of donor dsDNA is incorporated into site I to form a new heteroduplex with the invading ssDNA in the filament[8,17–19].

Site I, which orientates around the inside of the presynaptic filament, contains two conserved flexible loops, L1 and L2[20–24]. By solving the crystal structure of the RecA presynaptic filament, Chen et al. showed that the ssDNA is held by L1 and L2 along the central axis in site I, with its nucleotides exposed to the internal cavity[15]. Two hydrophobic amino acid residues, Met-164 in L1 and Ile-199 in L2, protrude into inter-triplet gaps, thereby stabilizing the elongated ssDNA. By contrast, in the postsynaptic filament, in which dsDNA is bound to site I, Met-164 in L1 inserts into the inter-triplet gap of the complementary strand, whereas Ile-199 remains in a stacking interaction with the initial ssDNA[15].

Site II is positioned closely parallel to site I along the presynaptic filament. It was proposed that the presynaptic filament makes contacts transiently and non-sequence-specifically with an incoming dsDNA via site II to form a nascent three-stranded synaptic joint[14,16]. If sufficient homology is found, the complementary strand of donor dsDNA is transferred to the recipient ssDNA of the presynaptic filament, resulting in heteroduplex formation. Arg-243 and Lys-245 of RecA are critical for site II function[25].

The structural features of the DNA binding sites in *E. coli* RecA are highly conserved among recombinases. Therefore, the fundamental processes of strand exchange driven by eukaryotic recombinases are likely to be very similar to those driven by RecA. Indeed, a cryo-electron microscopy (cryo-EM) study demonstrated that the near-atomic resolution structures of the human RAD51 (HsRAD51)-ssDNA and HsRAD51-dsDNA complexes, corresponding to the presynaptic and postsynaptic complexes, respectively, are very similar to the equivalent RecA structures[26]. The authors proposed that Val-273 in L2, which corresponds to Ile-199 of RecA, inserts into the inter-triplet gaps of ssDNA, thereby stabilizing the asymmetric ssDNA elongation. Val-273 also inserts into the inter-triplet gaps of dsDNA, suggesting that L2 stabilizes the heteroduplex DNA product during the DNA strand exchange reaction. In addition, Arg-235 in L1 inserts into the inter-triplet gaps and interacts with the phosphate backbone of one strand of the dsDNA. Intriguingly, RecA lacks the amino acid corresponding to Arg-235, implying that Arg-235 may exert a role that distinguishes the strand exchange reaction driven by eukaryotic recombinases from that driven by the prokaryotic recombinase RecA. Although these structural studies are consistent with the possibility that site I and II function as the catalytic core of RecA family recombinases, it is still unclear if this is the case for eukaryotic Rad51.

By developing a real-time monitoring assay, we recently showed that the strand exchange reaction driven by Rad51 from the fission yeast *Schizosaccharomyces pombe* (SpRad51) proceeds via two distinct three-stranded intermediates, complex 1 (C1) and complex 2 (C2)[27]. Thus, the reaction consists of three steps: formation of C1, transition from C1 to C2, and release of the non-complementary donor strand from C2. The C1 and C2 intermediates have different structural characteristics. The donor dsDNA retains the original base pairs in C1, whereas in C2 the initial ssDNA is intertwined with the complementary strand of the donor dsDNA. Therefore, C1 and C2 correspond closely to paranemic and plectonemic joints, respectively, which the Radding group originally proposed as intermediates of the RecA-driven DNA strand exchange reaction[28]. The Swi5-Sfr1 complex, a highly conserved Rad51/Dmc1 activator[29,30], strongly stimulates the second (C1–C2 transition) and the third (C2 to final product formation) steps of DNA strand exchange[27].

In this study, to elucidate in detail the molecular roles of DNA binding sites I and II in eukaryotic recombinases, we characterize three DNA binding mutants of SpRad51 using various methods, including a fluorescence resonance energy transfer (FRET)-based real-time strand exchange assay that we previously developed[27,31]. We find that an L1 mutant is defective in formation of C1, whereas an L2 mutant abrogates the C1–C2 transition that produces the heteroduplex. These results suggest that L1 and L2 play central but distinct catalytic roles in the DNA strand exchange reaction leading to heteroduplex formation. In addition, our findings suggest that site II serves as an entry gate to deliver not only dsDNA, but also ssDNA, to the catalytic site I, even though site II was previously thought to play the predominant role in donor dsDNA capture and homology search. Thus, our data provide several new insights into the common molecular mechanisms underlying the strand exchange reaction driven by eukaryotic recombinases.

## Results

**Design of three DNA binding site mutants**. Rad51 has two DNA binding sites, site I and site II (Fig. 1a). Site I consists of two loops, L1 and L2; site II is located C-terminal to L2. We generated

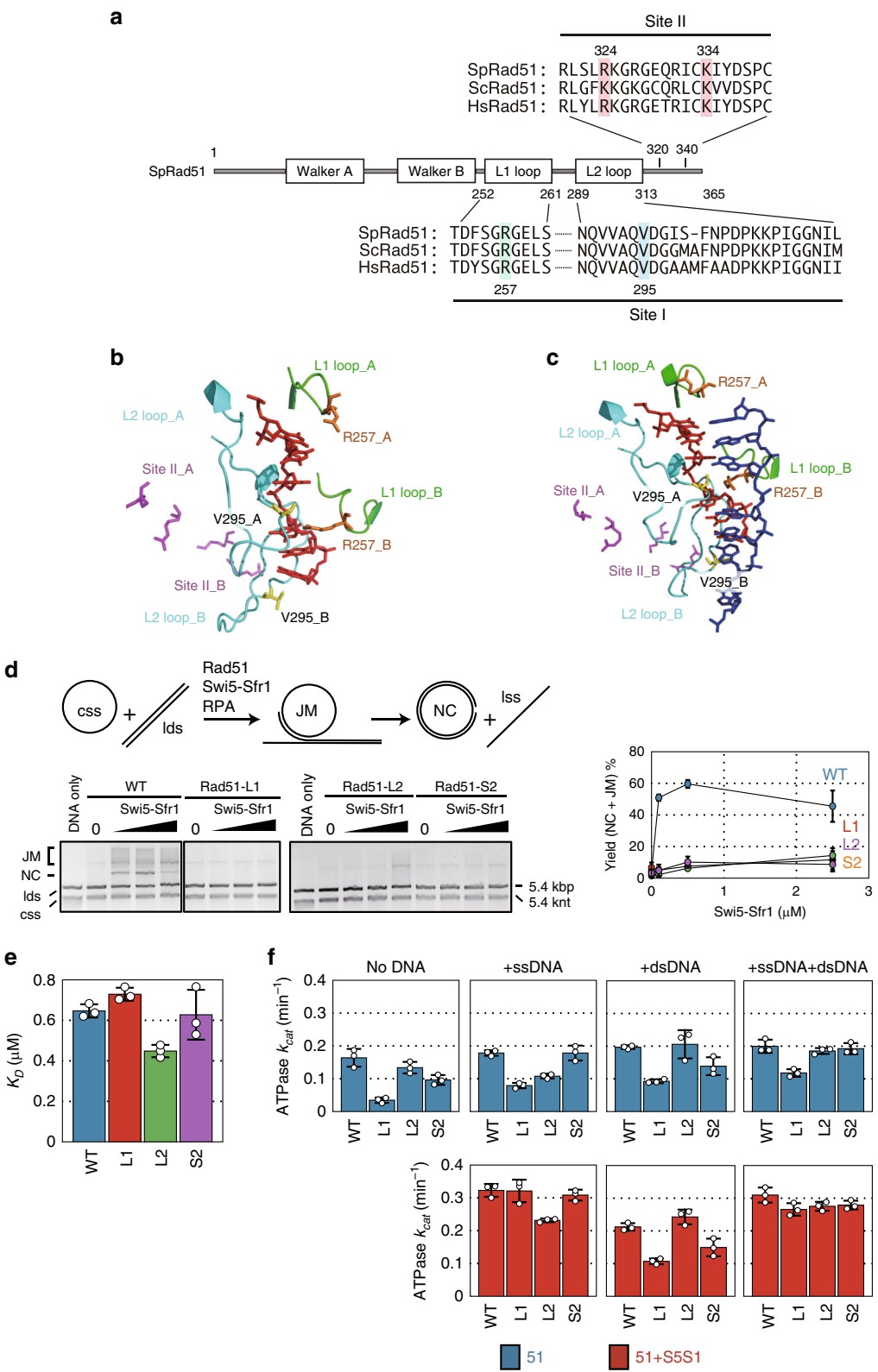

three mutants based on amino acid sequence conservation. The Rad51-L1 mutant has a single mutation in L1, Arg-257 to Ala (R257A); this corresponds to Arg-235 in HsRad51, which was proposed to be directly involved in dsDNA binding[26]. The Rad51-L2 mutant has a single mutation in L2, Val-295 to Ala (V295A), which corresponds to Val-273 in HsRad51 and Ile-199 in RecA. This residue has been proposed to insert into the inter-

triplet gap of ssDNA and dsDNA to stabilize the invading strand and the heteroduplex DNA product, respectively[15,26]. Finally, Rad51-S2, a site II mutant, contains two mutations, Arg-324 to Ala and Lys-334 to Ala (R324A and K334A); these residues correspond to Arg-243 and Lys-245, respectively, in site II of RecA[15,25]. Since these two basic amino acid residues have been proposed to play an equally important role in DNA binding, we

**Fig. 1 Rad51 DNA binding site mutants are defective in DNA strand exchange. a** Domain structures of *S. pombe* Rad51. DNA binding site I contains two loops, L1 and L2. Site II is located C-terminal to L2. Amino acid alignments of these regions are shown. Sp: *Schizosaccharomyces pombe*; Sc: *Saccharomyces cerevisiae*; Hs: *Homo sapiens*. Rad51 mutants with mutations R257A in L1, V295A in L2, and R324A K334A in Site II are referred to as Rad51-L1, Rad51-L2, and Rad51-S2, respectively. **b** A model of the SpRad51-ssDNA filament constructed by homology modeling using the HsRad51-ssDNA filament structure (PDBID:5H1B). The initial ssDNA is shown in red. L1 and L2 loops are in green and cyan, respectively. R257 in the L1 loop is shown in orange. V295 in the L2 loop is shown in yellow. R324 and K334 in site II are shown in magenta. **c** A model of the SpRad51-dsDNA filament constructed by homology modeling using the HsRad51-dsDNA filament structure (PDBID:5H1C). The initial ssDNA, L1 loop, L2 loop, R257 in L1 loop, V295 in L2 loop, and R324 and K334 in site II have the same colors as those in Fig. 1b. The complementary strand of the initial ssDNA is in blue. **d** DNA three-strand exchange assay using long DNA substrates. (upper left) Schematic of the assay. (lower left) Gel image of the three-strand exchange assay. Wild-type Rad51 (WT), Rad51-L1 (L1), Rad51-L2 (L2), or Rad51-S2 (S2) (5 μM each) and Swi5-Sfr1 (0.1, 0.5, and 2.5 μM) were used in the assay. (right) Quantified yields of nicked-circular dsDNA (NC, one of the final reaction products) and joint molecules (reaction intermediates). **e** $K_D$ values of wild-type and mutant Rad51 proteins for trinitrophenyl-ATP. WT, 0.646 ± 0.032; Rad51-L1, 0.729 ± 0.031; Rad51-L2, 0.448 ± 0.030; Rad51-S2, 0.627 ± 0.12. **f** $k_{cat}$ values of ATP hydrolysis by wild-type and mutant Rad51 proteins. Blue bars, Rad51 only. Red bars, Rad51 with Swi5-Sfr1. Data (**d–f**) are expressed as the mean ± s.d. (*n* = 3 independent experiments). Source data are provided as a Source data file.

simultaneously substituted both of them to Ala in the S2 mutant. The positions of these residues in the presynaptic (Fig. 1b) and postsynaptic (Fig. 1c) filaments of SpRad51 are shown. To analyze these Rad51 mutants in vitro, they were purified to homogeneity using an *E. coli* overexpression system (Supplementary Fig. 1a). The three mutants showed circular dichroism (CD) profiles and thermal stability that was comparable to wild-type Rad51, suggesting that the mutants have no obvious impairment in their overall structure (Supplementary Fig. 1b, c). In addition, the three mutants were able to bind the Swi5-Sfr1 auxiliary factor similarly to wild-type Rad51, as judged by a coimmunoprecipitation assay (Supplementary Fig. 1d).

**Rad51 DNA binding site mutants are severely defective in DNA strand exchange in vitro.** We first examined the in vitro DNA strand exchange activity of these mutant Rad51 proteins. For this purpose, we employed a conventional three-strand exchange assay using ϕX174 viral circular ssDNA (cssDNA) and *ApaL*I-linearized dsDNA (ldsDNA) as substrates (Fig. 1d). Pairing between cssDNA and ldsDNA yields DNA joint molecules (JM) as intermediates, and DNA strand exchange over the 5.4-kb length of the synapsed substrates yields nicked circular duplex (NC) and linear ssDNA as products. When reactions containing wild-type Rad51 were supplemented with Swi5-Sfr1, robust production of JM and NC was observed. In sharp contrast, these DNA species were barely detectable when mutant Rad51 proteins were employed, indicating that all three mutants have substantially reduced strand exchange activity.

**ATP binding and hydrolysis by the Rad51 mutants.** Since ATP binding and hydrolysis are important for Rad51-driven DNA strand exchange, we first examined these activities of the Rad51 mutants (Fig. 1e, f and Supplementary Table 1). The dissociation constants ($K_D$) of the L1 and S2 mutants for trinitrophenyl-ATP (TNP-ATP), a fluorescent analog of ATP, were comparable to wild-type Rad51 (Fig. 1e), indicating normal ATP binding. The $K_D$ values of the L2 mutant is slightly lower than that of wild-type Rad51, indicating a slightly higher affinity for ATP. A reduction in the ATPase activity of the L1 mutant was observed under most conditions in which cofactors (i.e., Swi5-Sfr1 and DNA) were omitted (Fig. 1f). In reactions that were only supplemented with ssDNA, the L2 mutant also showed a reduction in ATP hydrolysis. The ATPase activity of the S2 mutant was also slightly reduced in the absence of cofactors and in the presence of dsDNA only. Despite these apparent deficiencies, all three mutants showed wild-type levels of ATP hydrolysis in the presence of ssDNA, dsDNA and Swi5-Sfr1 (i.e., under the conditions employed in the strand exchange assay), indicating that the near-

complete loss of strand exchange activity in these mutants (Fig. 1d) cannot be explained by a defect in ATP binding or hydrolysis.

**FRET-based real-time analysis of DNA strand exchange activity.** To determine which step in the DNA strand exchange reaction is defective in these Rad51 mutants, we performed FRET-based real-time assays to monitor DNA strand pairing and displacement[27] (Fig. 2). In the pairing assay, the Rad51 presynaptic filament is formed on fluorescein-labeled ssDNA (83 nt) and then mixed with rhodamine-labeled dsDNA (40 bp). Once the presynaptic filament interacts with dsDNA, a reaction intermediate containing three-stranded DNA is formed (C1), and this is then processed into another intermediate (C2). The C1–C2 transition is followed by displacement of the unlabeled strand of dsDNA, which culminates in heteroduplex formation. Upon formation of C1, the first of the two intermediates, the fluorescence emission of fluorescein decreases because rhodamine quenches the emission by FRET (Fig. 2a). By contrast, the ssDNA in the Rad51 filament is unlabeled in the DNA displacement assay. Instead, the 5′-end of one strand of the homologous donor dsDNA is labeled with fluorescein and the 3′-end of its complementary strand is labeled with rhodamine. Once the presynaptic filament interacts with the homologous dsDNA, the C1 intermediate is formed and processed into the C2 intermediate, which is then followed by displacement of the fluorescein-labeled ssDNA from the donor dsDNA. Before the presynaptic strand is added, the two dyes on dsDNA can undergo FRET. After DNA displacement from the C2 intermediate occurs, fluorescein is freed from rhodamine, increasing its fluorescence emission (Fig. 2b). Importantly, Rad51 does not affect fluorescence emission of fluorescein or FRET efficiency of fluorescein by rhodamine in either assay[27].

Consistent with our previous work[27], wild-type Rad51 converted ~20% substrates into intermediates within 40 min in the pairing assay (Fig. 2a). Swi5-Sfr1 stimulated Rad51 activity by ~2-fold. In the DNA displacement assay, wild-type Rad51 converted 10% of substrates into the final products within 40 min (Fig. 2b). Swi5-Sfr1 increased the displacement activity of wild-type Rad51 protein by ~4-fold. To analyze the reaction kinetics of strand exchange, we simulated the pairing reaction of wild-type Rad51 using DynaFit software ver. 4.08.016[32] and obtained its reaction constants (Fig. 2c and Supplementary Table 2). Residuals between experimental data of the DNA strand pairing assay and a theoretical curve obtained by simulation using DynaFit indicated a very good fit (Supplementary Fig. 2). These results confirmed that the DNA strand exchange reaction mediated by Rad51 obeys the sequential three-step reaction model involving two distinct three-stranded intermediates, C1

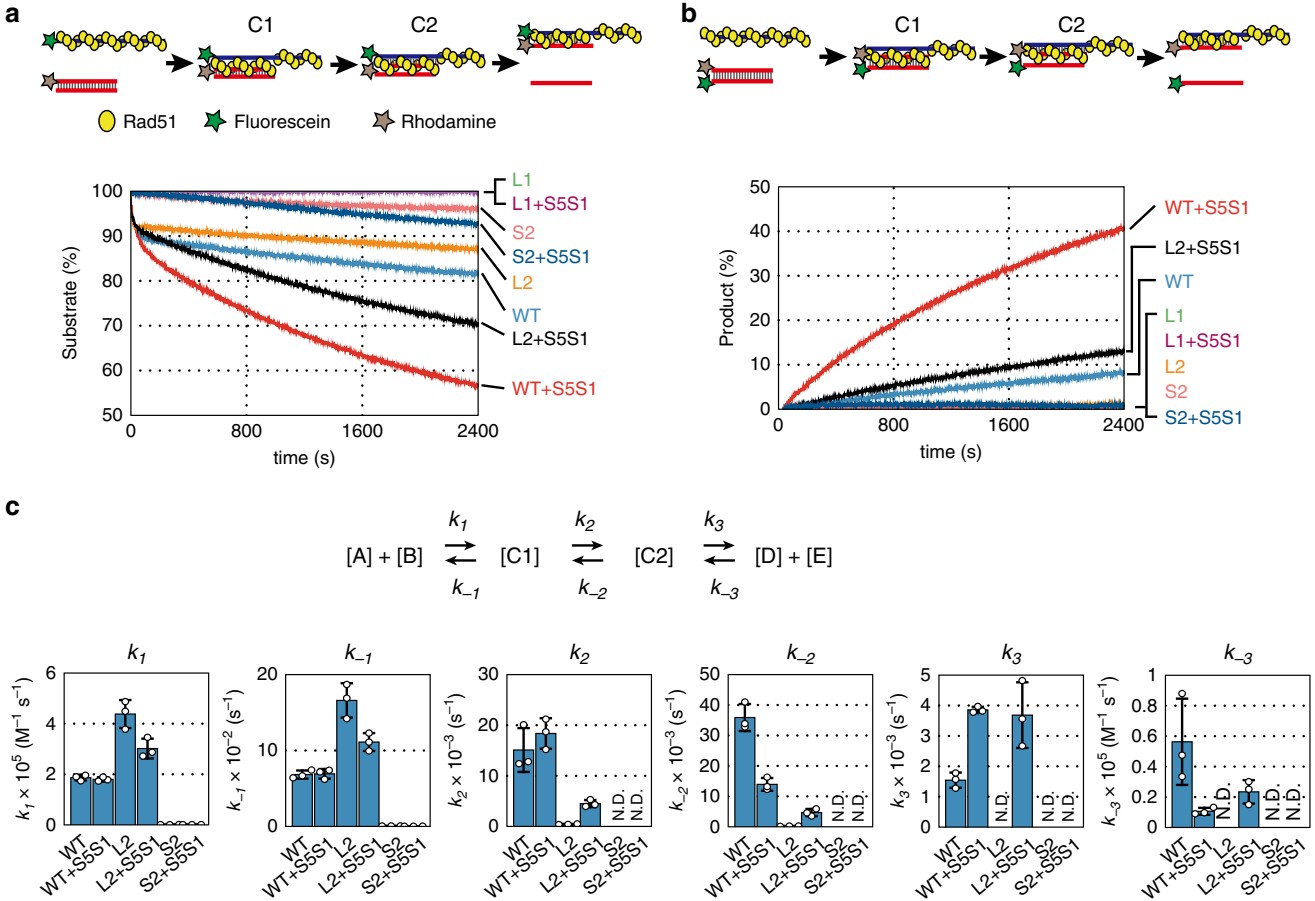

**Fig. 2 DNA strand exchange in real time. a** FRET-based real-time assays for the Rad51-mediated DNA strand pairing reaction. (upper) Schematic diagram of the pairing reaction. Yellow circles represent Rad51 monomers. Green and brown stars represent fluorescein and rhodamine, respectively. (lower) Time course of the DNA strand pairing reaction. Reactions containing fluorescein-labeled presynaptic filaments (83 nt) consisting of wild-type Rad51, Rad51-L1 (L1), Rad51-L2 (L2), or Rad51-S2 (S2) (1.5 μM each) with/without Swi5-Sfr1 (0.15 μM) were started (time = 0) by addition of donor dsDNA (40 bp), which was labeled with rhodamine at the end of the complementary strand of the presynaptic filament. Fluorescence emission was monitored at 525 nm upon excitation at 493 nm. Wild-type Rad51; light blue, wild-type Rad51 with Swi5-Sfr1; red, Rad51-L1; green, Rad51-L1 with Swi5-Sfr1; purple, Rad51-L2; yellow, Rad51-L2 with Swi5-Sfr1; black, Rad51-S2; orange, Rad51-S2 with Swi5-Sfr1; dark blue. **b** FRET-based real-time assays of the Rad51-mediated DNA strand displacement reaction. (upper) Schematic diagram of the displacement reaction. (lower) Time course of the displacement reaction. Reaction conditions were the same as in (**a**), except that the two ends of the donor dsDNA were labeled with fluorescein and rhodamine. **c** The upper panel shows the schematics of the three-step reaction. In the reaction formula, [A], [B], [C1], and [C2] correspond to the concentrations of presynaptic filament, donor dsDNA, C1 intermediate, and C2 intermediate, respectively. [D] and [E] correspond to the two products of the strand exchange reaction, heteroduplex DNA, and released ssDNA, respectively. The lower panel shows rate constants for DNA strand pairing derived from the DynaFit-based fitting procedure applied to the data obtained in (**a**). Data are expressed as the mean ± s.d. ($n = 3$ independent experiments). See also Supplementary Table 2. Source data are provided as a Source data file.

and C2, and that Swi5-Sfr1 stimulates the C1–C2 transition and the final DNA displacement step. The reaction curve showing phasic variation in the time-dependent signal is consistent with this notion.

**Mutation of the L1 loop abolishes both pairing and displacement activities of Rad51.** The L1 mutant exhibited neither pairing nor displacement activity, even in the presence of Swi5-Sfr1 (Fig. 2a, b). Consequently, we could not calculate kinetic values for these reactions. This result indicates that the R257A mutation completely abolishes the DNA strand exchange activity of Rad51.

**Mutation of the L2 loop hinders C1 progression.** The Rad51-L2 loop mutant exhibited a very similar curve to that of wild-type Rad51 within the initial 50 s of the pairing assay. However, after this point, the progression of pairing showed a marked decline

(Fig. 2a). Moreover, Rad51-L2 did not exhibit any displacement activity (Fig. 2b). The Swi5-Sfr1 complex stimulated both the pairing and displacement activities of Rad51-L2, but only ~40% of the intermediates were converted into final products by 40 min (despite ~30% of the substrates forming intermediates in the pairing assay, only ~12% of the substrates yielded the final product in the displacement assay). By contrast, wild-type Rad51 was able to convert ~95% of intermediates into final products (~43% of substrates formed intermediates in the pairing assay and ~41% of intermediates yielded final products in the displacement assay). These results suggest that Rad51-L2 can form reaction intermediates but cannot efficiently process these intermediates into the final products. The pattern of the reaction curve with a very steep slope at the early time point and a drastically reduced slope at a later time also supports this notion.

To analyze the reaction kinetics of strand exchange driven by Rad51-L2, we also simulated the pairing reaction of Rad51-L2

(Fig. 2c and Supplementary Table 2). As for wild-type Rad51, the residuals between experimental data for Rad51-L2 and a theoretical curve obtained by simulation indicated a very good fit (Supplementary Fig. 2). Both the $k_1$ and $k_{-1}$ values of the mutant were ~2.4-fold higher than those of wild-type Rad51, resulting in similar $K_1$ ($k_1$ per $k_{-1}$) values. This indicates that, although the amounts of C1 in the wild-type Rad51 and Rad51-L2 reactions were comparable, formation and disruption of C1 occurred more frequently in reactions mediated by Rad51-L2 than those mediated by wild-type Rad51. By contrast, although the $K_2$ ($k_2$ per $k_{-2}$) value of the mutant was ~5-fold higher than that of wild-type Rad51, the $k_2$ and $k_{-2}$ values of Rad51-L2 were ~36 and ~165-fold smaller, respectively, than those of wild-type Rad51. These results clearly indicate that the dynamism of the C1–C2 transition is reduced in the Rad51-L2 mutant. The third step was too slow for a confident determination of the reaction rate constants $k_3$ and $k_{-3}$.

Swi5-Sfr1 increased the $K_2$ and $K_3$ ($k_3$ per $k_{-3}$) values of the reaction catalyzed by wild-type Rad51, mainly via a ~2.5-fold decrease in $k_{-2}$, a ~2.5-fold increase in $k_3$, and a ~5.3-fold decrease in $k_{-3}$ (Supplementary Table 2), as reported previously[27]. Although Swi5-Sfr1 also increased the $k_2$ and $k_{-2}$ values of the Rad51-L2 reaction (~10- and ~21-fold, respectively), the resultant values were still ~4.1- and ~3.0-fold less, respectively, than those for wild-type Rad51 (Fig. 2c). Swi5-Sfr1 increased $k_3$ and $k_{-3}$ of the reaction catalyzed by Rad51-L2 to values similar to those of wild-type Rad51.

Taken together, these findings indicate that Rad51-L2 is proficient for C1 formation, and despite retaining a functional interaction with Swi5-Sfr1, has a severe defect specifically in processing the C1–C2 transition.

**Mutations in site II abolish C1 formation.** The pairing assay demonstrated that the Rad51-S2 reaction progressed much more slowly than the wild-type Rad51 reaction, and that Swi5-Sfr1 did not appreciably stimulate the pairing activity of Rad51-S2 (Fig. 2a). In addition, Rad51-S2 did not exhibit any displacement activity even in the presence of Swi5-Sfr1 (Fig. 2b). The residuals between experimental data and a theoretical curve obtained by simulation indicate a very good fit for Rad51-S2, despite the fact that the reaction progressed very slowly (Supplementary Fig. 2). Compared to wild-type Rad51, the $k_1$ and $k_{-1}$ values of the Rad51-S2 reaction were extremely low (~180- and ~108-fold less, respectively), even in the presence of Swi5-Sfr1 (~171- and ~240-fold less, respectively) (Fig. 2c and Supplementary Table 2). As a result, the progress of strand exchange reactions driven by Rad51-S2 was too slow to determine the reaction rate constants of later steps ($k_2$, $k_{-2}$, $k_3$, and $k_{-3}$). Thus, we conclude that Rad51-S2 is defective in C1 formation.

**C1 intermediates accumulate in the L2 loop mutant.** Based on the results presented in Fig. 2, we predicted that C1 intermediates would accumulate in the strand exchange reaction mediated by Rad51-L2. To directly test this, we carried out a pairing-initiated abortive DNA strand exchange assay (Fig. 3a)[27]. In this assay, EDTA is added to the pairing reactions 5, 10, or 20 min after the initiation of pairing to chelate $Mg^{2+}$, leading to nucleotide depletion and consequently dissociation of Rad51 from DNA, which in turn induces collapse of strand exchange intermediates. An increase in fluorescence emission is observed if intermediates collapse into substrates, which occurs if C1 intermediates accumulate because they cannot be converted into C2 intermediates. By contrast, fluorescence emission does not change substantially if intermediates collapse into products, which occurs if C2 intermediates (and reaction products) are more abundant than

C1. The results revealed that fluorescence emission increased upon EDTA addition at all time points, indicating that C1 intermediates had accumulated in all reactions tested (Fig. 3b–e).

To estimate the amount of the C1 intermediate that was converted into substrate, we calculated the change in fluorescence by subtracting the intensity before addition of EDTA from the intensity 200 s after addition of EDTA (Fig. 3f). In the reaction with wild-type Rad51 protein, ~50% of the reaction intermediates were converted into substrates following addition of EDTA. The Swi5-Sfr1 complex decreased the amount of C1, consistent with the notion that Swi5-Sfr1 stimulates the C1–C2 transition, thereby reducing the amount of C1 that would be available to collapse into substrates upon addition of EDTA. Rad51-L2 had accumulated ~2-fold more C1 intermediates than wild-type Rad51 by 20 min after initiation of the reaction (Fig. 3d, f). Although the inclusion of Swi5-Sfr1 resulted in a ~2-fold reduction in C1, this is still less than the ~4-fold reduction observed when wild-type Rad51 was combined with Swi5-Sfr1 (Fig. 3e, f). These results suggest that Rad51-L2 is defective in forming the C2 intermediate containing de novo heteroduplex DNA.

**Rad51-L2 is deficient in the formation of de novo heteroduplex DNA.** To more directly monitor the formation of heteroduplex DNA in C2, we adapted our strand exchange assay such that the ssDNA contained 2-aminopurine (2AP), a fluorescent analog of adenine that base-pairs with thymine[33–39]. The fluorescence emission of 2AP decreases upon 2AP-thymine base-pair formation, which should only occur upon formation of the C2 intermediate, reflecting heteroduplex formation by strand exchange[40,41] (Fig. 4a).

When wild-type Rad51, ssDNA containing 2AP, and dsDNA homologous to the ssDNA were combined, ~8% of substrates were converted into products (Fig. 4b, ho). Swi5-Sfr1 stimulated the reaction ~2-fold, consistent with our previous report[27]. In contrast, when the dsDNA was heterologous to the ssDNA, the change in emission was negligible (~10% of the reactions with homologous dsDNA) (Fig. 4b, he).

It is important to note that the FRET-based pairing assay described in Fig. 2 does not detect C2 formation per se. This is because C1 formation, which is sufficient for FRET, is not dependent on the existence of homology between the presynaptic filament and dsDNA (Fig. 4b). By contrast, the assay containing 2-AP only detects C2 formation, since a reduction in the substrate was only observed under conditions where heteroduplex DNA can form (i.e., when homologous substrates were employed) (Fig. 4b). Consistently, the 2-AP reactions containing homologous substrates and wild-type Rad51 (with or without Swi5-Sfr1) showed a quasi-linear curve, in contrast to the curve seen in the equivalent FRET-based pairing assays (Fig. 4b). Taken together, we conclude that this assay can indeed detect de novo heteroduplex DNA formation between homologous substrates.

When Rad51-L2 was employed in the 2-AP assay, the change in the substrate was ~2.9-fold less than that seen with wild-type Rad51 protein (Fig. 4c). Although the Swi5-Sfr1 complex stimulated the Rad51-L2 reaction, the amount of the product was ~1.7-fold less than the equivalent reaction containing wild-type Rad51 (Fig. 4d). Taken together, this finding further argues that Rad51-L2 is specifically defective in converting the C1 intermediate into the C2 intermediate, which contains de novo heteroduplex DNA formed by the strand exchange reaction.

**Properties of presynaptic filaments formed by the mutant Rad51 proteins.** Since the L1 and S2 mutants had clear defects in DNA pairing (Fig. 2a), we reasoned that these mutants may form

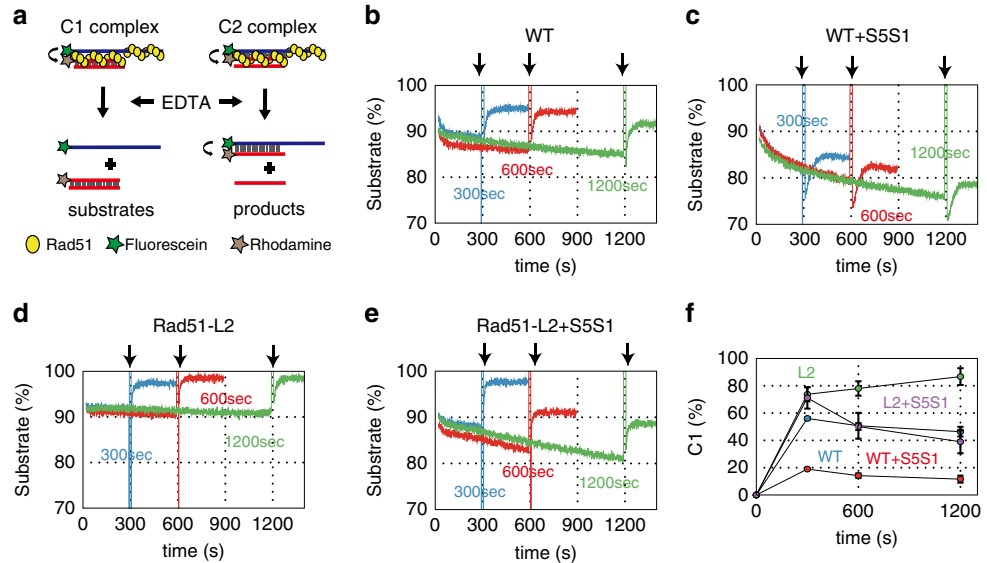

**Fig. 3 A pairing-initiated abortive DNA strand exchange assay indicates that C1 intermediates accumulate in the L2 loop mutant. a** Schematic diagram of a pairing-initiated abortive DNA strand exchange assay. EDTA was added to the ongoing pairing reactions to chelate $Mg^{2+}$, leading to dissociation of Rad51 from the intermediates. Fluorescence emission increases if the C1 intermediate is converted to substrates, but does not change substantially if the C2 intermediates are converted to products. **b–e** Time courses of the pairing-initiated abortive assay: **b** wild-type Rad51; **c** wild-type Rad51 with Swi5-Sfr1; **d** Rad51-L2; and **e** Rad51-L2 with Swi5-Sfr1. EDTA was added in the reaction mixture 5 (blue), 10 (red), or 20 (green) min, as indicated by arrows, after the strand pairing reaction started. **f** Time course of accumulation (% of the input) of the C1 intermediate. wild-type Rad51; blue, wild-type Rad51 with Swi5-Sfr1; red, Rad51-L2; green and Rad51-L2 with Swi5-Sfr1; purple. Data are expressed as the mean ± s.d. ($n = 3$ independent experiments). Source data are provided as a Source data file.

aberrant presynaptic filaments. By contrast, if the defects of the L2 mutant are limited to the C1–C2 transition, then it should be able to form presynaptic filaments that are comparable to wild type. Therefore, we tested the steady-state ssDNA binding activity of the Rad51 mutants using fluorescence anisotropy. Consistent with these expectations, the L1 and S2 mutants exhibited a mild defect in ssDNA binding activity with slightly higher $K_D$ values (1.8- and 2.5-fold, respectively), whereas the L2 mutant had ssDNA binding activity comparable with that of wild-type Rad51 (Fig. 5a and Supplementary Table 3). Consistently, analysis of ssDNA binding via electrophoresis mobility shift assays (EMSA; protein-ssDNA complexes were covalently cross-linked to preserve labile structures prior to electrophoresis) demonstrated that the L2 mutant showed a similar mobility shift to wild-type Rad51, while the L1 and S2 mutants exhibited a slightly reduced mobility shift than wild-type Rad51 (Supplementary Fig. 3a).

To further characterize the ssDNA binding activity of these Rad51 mutants, we analyzed association with ssDNA in real time (Fig. 5b). At a concentration of 0.4 μM protein, the L1 and S2 mutants did not elicit a change in anisotropy to the same degree as the wild-type protein. In contrast, reactions containing the L2 mutant had the same anisotropy as wild-type Rad51. Thus, the L2 mutant has normal ssDNA binding activity whereas the L1 and S2 mutants are severely deficient in association with ssDNA, consistent with the result of the steady-state binding assays (Fig. 5a). When ssDNA and Rad51-L1 were added at two-fold higher concentrations than those of the standard condition, the L1 mutant showed similar association kinetics to that of the wild-type protein, suggesting that the L1 mutant has nearly normal ssDNA association potential (Fig. 5b). On the other hand, when ssDNA and Rad51-S2 were added at three-fold higher concentrations than those of the standard condition, the total increase in anisotropy was similar to wild-type Rad51 but the rate of increase was much slower, suggesting that Rad51-S2 is severely defective in association with ssDNA (Fig. 5b). Under these experimental conditions, the effect of Swi5-Sfr1 on ssDNA binding was

marginal for wild-type Rad51 and the mutants (Supplementary Fig. 3b).

Next, to examine the dissociation of presynaptic filaments formed by the mutants, we diluted (1:40) the reaction mixtures of the association assays containing the Rad51-ssDNA filaments into reaction buffer without any protein or DNA and monitored filament dissociation in real time. The dissociation rate constants ($k_{off}$) of these mutants for ssDNA were calculated by the first order decay model (Fig. 5c, Supplementary Fig. 4, Supplementary Table 4). The $k_{off}$ value of wild-type Rad51 was 0.0116 ($s^{-1}$). The presence of 0.04 μM and 0.2 μM of Swi5-Sfr1 reduced the $k_{off}$ values by ~4.3- and ~9-fold, respectively, indicating that Swi5-Sfr1 stabilizes the Rad51 filament by preventing the dissociation of Rad51 from ssDNA, consistent with previous reports[42,43]. In the absence of Swi5-Sfr1, filaments formed with the L2 mutant displayed a $k_{off}$ value similar to wild-type Rad51, while the L1 and S2 mutants had slightly higher $k_{off}$ values (Fig. 5c). The addition of Swi5-Sfr1 led to a marked reduction in the $k_{off}$ values for all three mutants, albeit to a lesser extent for the L2 mutant. These results indicate that the filaments formed by the L1 and S2 mutants are efficiently stabilized by Swi5-Sfr1. The apparent reduction in Rad51-L2 filament stability may be inconsequential since it formed a similar amount of the C1 intermediate as wild-type Rad51 (Fig. 2). Notably, these results indicate that the low affinity of the S2 mutant for ssDNA is not caused by rapid dissociation from ssDNA, but rather by slow association with ssDNA, suggesting that site II is involved in ssDNA capture.

**Rad51 presynaptic filament elongation is only mildly defective in the L2 mutant**. The ssDNA in the active form of the presynaptic filament is about 1.5× longer than B-form DNA[26,44,45]. To determine whether the mutant Rad51 proteins under examination here formed elongated presynaptic filaments, we utilized another FRET-based assay (Fig. 5d). In this assay, 73-mer ssDNA internally labeled with both fluorescein and rhodamine (separated

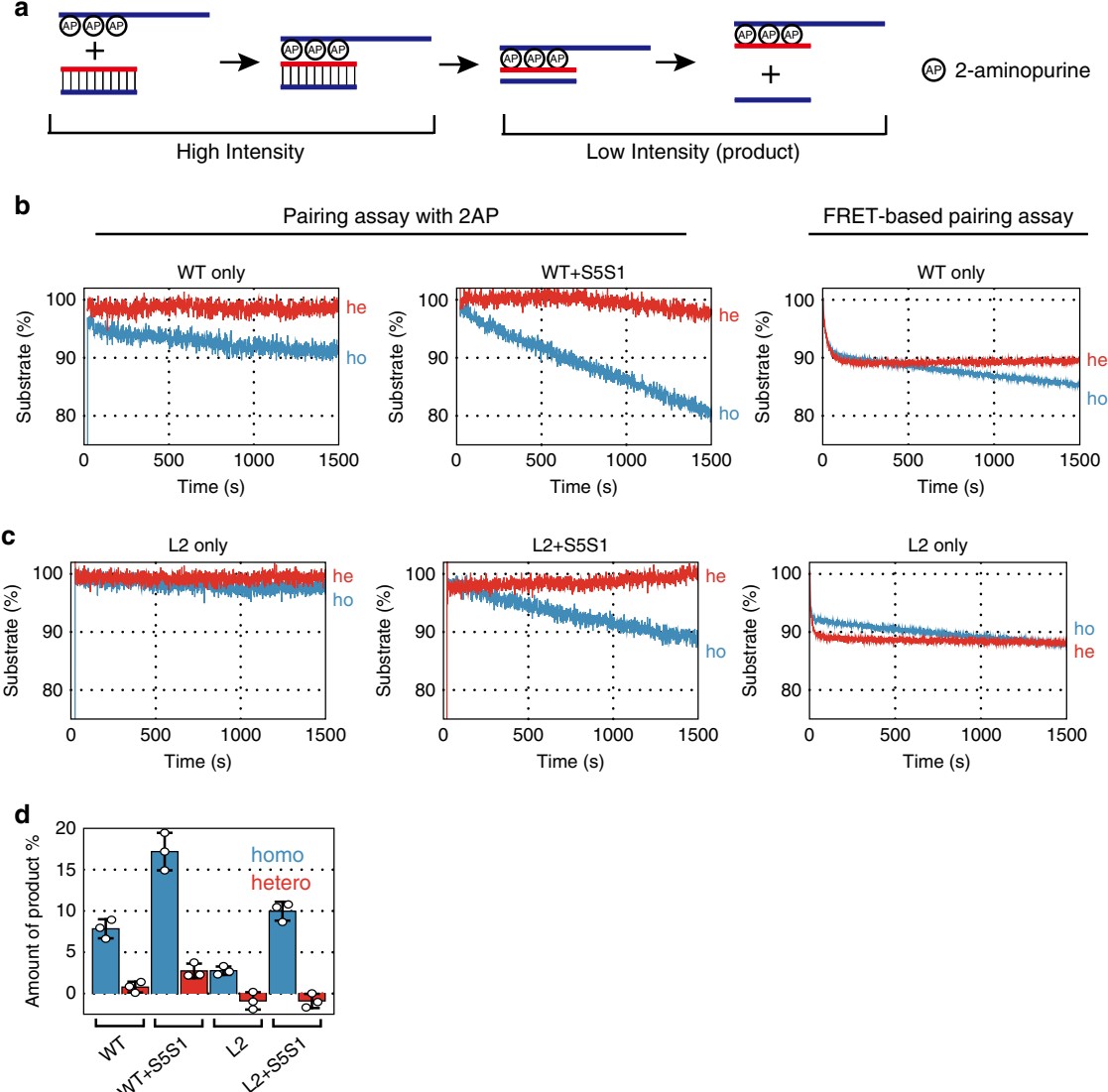

**Fig. 4 A pairing assay with 2AP indicates that Rad51-L2 is defective in forming the C2 intermediate, which contains heteroduplex DNA. a** Schematic diagram of DNA strand exchange reaction using ssDNA containing 2AP. **b** Time course of the DNA strand pairing reaction using wild-type Rad51. (left) A pairing reaction by Rad51 with 2AP-containing ssDNA and dsDNA with homologous (blue) or heterologous (red) sequences. (center) A pairing reaction by Rad51 + Swi5-Sfr1 with 2AP-containing ssDNA and dsDNA with homologous (blue) or heterologous (red) sequences. (right) Time course of the FRET-based DNA strand pairing reaction using wild type Rad51, fluorescein-labeled ssDNA, and rhodamine-labeled dsDNA with homologous (blue) or heterologous (red) sequences. **c** Time course of the DNA strand pairing reaction using Rad51-L2 mutant. (left) A pairing reaction by Rad51-L2 with 2AP-containing ssDNA and dsDNA with homologous (blue) or heterologous (red) sequences. (center) A pairing reaction by Rad51-L2 + Swi5-Sfr1 with 2AP-containing ssDNA and dsDNA with homologous (blue) or heterologous (red) sequences. (right) Time course of the FRET-based DNA strand pairing reaction using Rad51-L2, fluorescein-labeled ssDNA, and rhodamine-labeled dsDNA with homologous (blue) or heterologous (red) sequences. **d** Change in fluorescence intensity of 2AP 25 min after the pairing reaction started, calculated from (**b**). The results of reactions with homologous dsDNA and completely heterologous dsDNA are displayed as blue and red bars, respectively. Data are expressed as the mean ± s.d. ($n = 3$ independent experiments). Source data are provided as a Source data file.

by 11 nucleotides) was employed as the substrate for presynaptic filament formation. The binding of Rad51 to the ssDNA is expected to linearize the ssDNA, effectively increasing the distance between fluorescein and rhodamine and thus reducing the FRET efficiency ($E_{FRET}$). Consistently, when the double-labeled ssDNA was mixed with wild-type Rad51, $E_{FRET}$ decreased by ~50% (Fig. 5d and Supplementary Table 5), which corresponds to 1.45× the length of B-form DNA. The $E_{FRET}$ value decreased further in the presence of Swi5-Sfr1 (1.58× B-form DNA). This result is consistent with previous reports showing that Swi5-Sfr1 elongates the presynaptic filament and maintains it in an active state[46,47]. When ATP was omitted from the reaction, this

elongation was largely impaired (Supplementary Fig. 5a). AMP-PNP, a nonhydrolyzable ATP analog, also induced similar elongation (1.58× B-form DNA)(Fig. 5d, Supplementary Fig. 5a, b and Supplementary Table 5). These results indicate that ATP binding without subsequent hydrolysis is sufficient for Rad51 filament elongation. The L1 and S2 mutants formed elongated filaments comparable to those of wild-type Rad51, and exhibited similar responses to Swi5-Sfr1 (Fig. 5d, Supplementary Fig. 5a, b and Supplementary Table 5). The Rad51-L2 mutant showed a slight defect in filament elongation in the absence of Swi5-Sfr1 (1.40x B-form DNA compared to 1.45x for wild type), but no defect was observed in the presence of the activator.

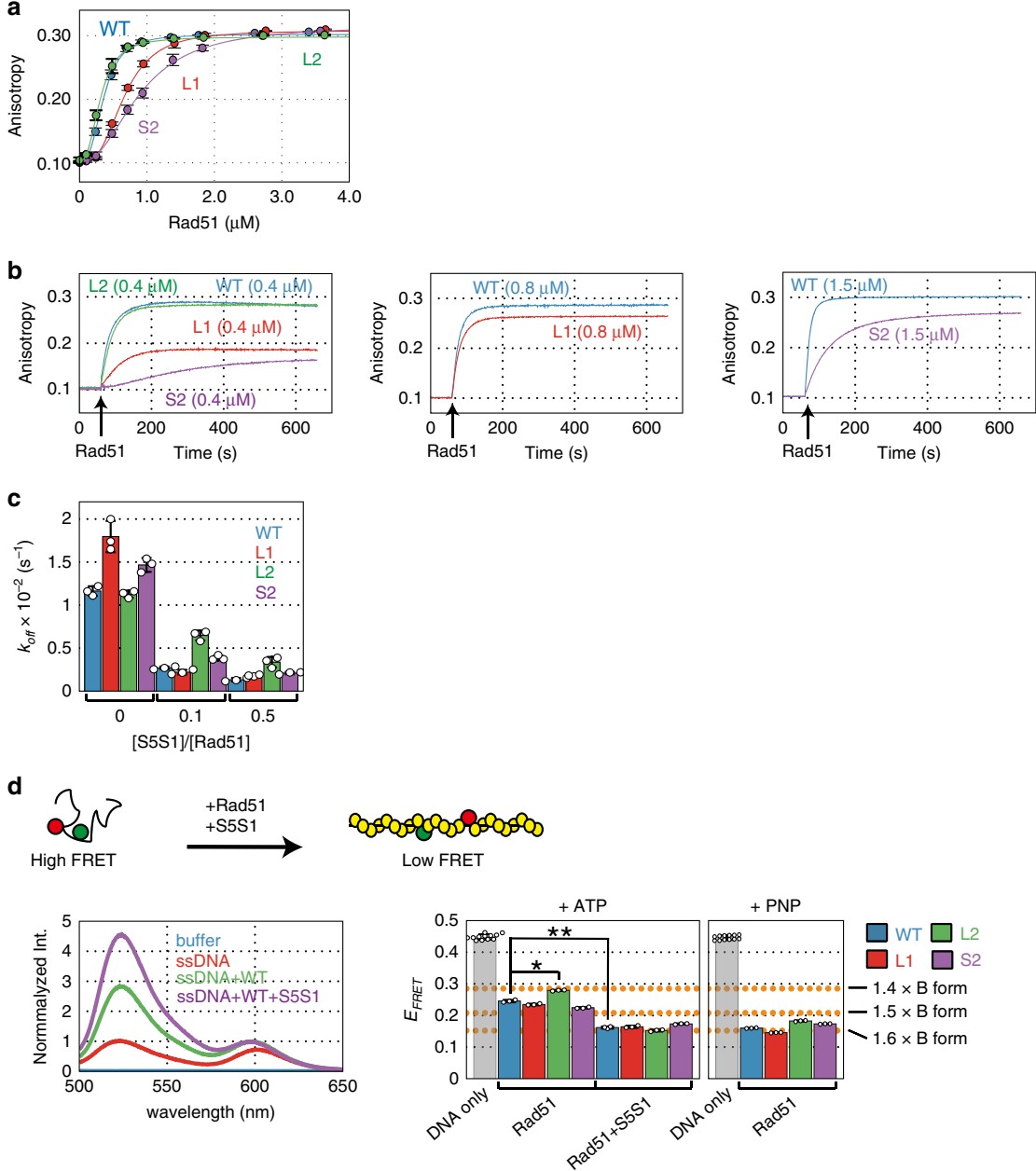

**Fig. 5 Formation and elongation of presynaptic filaments formed by Rad51 DNA binding site mutants. a** Formation of presynaptic filaments by wild-type and mutant Rad51 proteins was observed by measuring fluorescence anisotropy. Rad51 protein was titrated into the reaction mixture containing ssDNA (3 μM nucleotide). Wild-type Rad51; blue, Rad51-L1; red, e, Rad51-L2; green, Rad51-S2; purple. **b** Kinetics of the association between Rad51 and ssDNA. Wild-type or mutant Rad51 protein was injected into a cuvette containing TAMRA-labeled ssDNA 60 s after measurement of fluorescence anisotropy started, as described in "Methods". (left) Association kinetics of wild-type Rad51 (blue), Rad51-L1 (red), Rad51-L2 (green), and Rad51-S2 (purple) (0.4 μM each) on TAMRA-labeled ssDNA (1.2 μM). (center) Association kinetics of wild-type Rad51 and Rad51-L1 (0.8 μM each) on TAMRA-labeled ssDNA (2.4 μM). (right) Association kinetics of wild-type Rad51 and Rad51-S2 (1.5 μM) on TAMRA-labeled ssDNA (4.5 μM). **c** Dissociation rate constants of the DNA binding mutants from ssDNA, calculated from the dissociation assays in Supplementary Fig. 4. **d** Elongation of the presynaptic filament. (upper) Schematic diagram depicting observation of the elongation of the presynaptic filament using FRET. Yellow balls represent Rad51 monomers. Green and red balls represent fluorescein and rhodamine, respectively. (lower left) Emission spectra of fluorescein and rhodamine, which are separated by 11 nucleotides, collected by excitation at 493 nm. Wild-type Rad51 (3 μM) and Swi5-Sfr1 (0.3 μM) were mixed with double-labeled ssDNA (3 μM, nucleotide) in the presence of ATP. (lower right) FRET efficiency ($E_{FRET}$) in the presence of ATP or AMP-PNP was calculated from the emission spectra shown here and in Supplementary Fig. 5. Orange dot-lines show the distances between two fluorophores compared with B-form DNA. *$p = 1.53 \times 10^{-4}$ and **$p = 8.64 \times 10^{-6}$, by two-tailed Student's $t$ test. Data (**a**, **c**, **d**) are expressed as the mean ± s.d. ($n = 3$ independent experiments, except $n = 12$ for DNA only in **d**). Source data are provided as a Source data file.

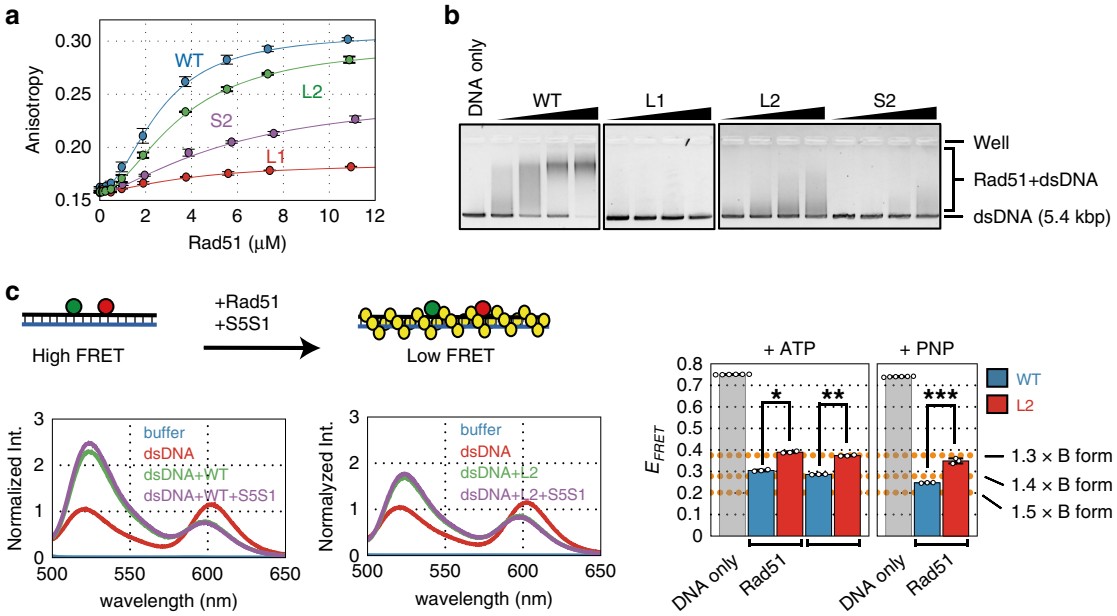

**Fig. 6 Rad51-L1 and -S2 are defecting in dsDNA binding, whereas Rad51-L2 binds but does not completely elongate dsDNA. a** Formation of Rad51-dsDNA filaments by wild-type Rad51 (blue), Rad51-L1 (red), -L2 (green), or -S2 (purple) was observed by measuring fluorescence anisotropy. Rad51 protein was titrated into the reaction mixture containing dsDNA (3 μM bp). **b** Gel shift assay. Various concentrations of wild-type and mutant Rad51 proteins (1.25, 2.5, 5.0, and 10 μM) were mixed with dsDNA (10 μM nucleotide). After incubation at 37 °C for 15 min, Rad51-dsDNA filaments were cross-linked with glutaraldehyde. **c** Elongation of Rad51-dsDNA filament. (upper) Schematic diagram depicting observation of the elongation of Rad51-dsDNA filament using FRET. Emission spectra of fluorescein (green) and rhodamine (red), which are separated by 11 nucleotides on the same strand, were collected by excitation at 493 nm. Yellow balls represent Rad51 monomers. (lower) Emission spectra of fluorescein and rhodamine. Wild-type Rad51 (lower left) or Rad51-L2 (lower center) (8 μM each) and Swi5-Sfr1 (0.8 μM) were mixed with the double-labeled dsDNA (3 μM bp) and the emission spectra of fluorescein and rhodamine were collected after excitation at 493 nm, as described in the "Methods". (lower right) FRET efficiency ($E_{FRET}$) of each reaction condition containing wild-type Rad51 or Rad51-L2 in the presence of ATP or AMP-PNP was calculated from the emission spectra shown here and in Supplementary Fig. 5. Orange dot-lines in **d**, **e** show the distances between two fluorophores compared with B-form DNA. *$p = 1.32 \times 10^{-5}$, **$p = 4.17 \times 10^{-13}$ and ***$p = 1.17 \times 10^{-5}$ by two-tailed Student's $t$ test. Data (**a**, **c**) are expressed as the mean ± s.d. ($n = 3$ independent experiments, except $n = 6$ independent experiments for DNA only in **c**). Source data are provided as a Source data file.

Collectively, these ssDNA binding experiments reveal that, although the L1 and S2 mutants have some defects in ssDNA association, once the filaments are formed by the two Rad51 mutants, they do not exhibit any significant defects. The L2 mutant has no significant defect in ssDNA binding and only a marginal defect in ssDNA elongation in the absence of Swi5-Sfr1. Therefore, we surmise that the deficiency responsible for impairing DNA strand exchange by Rad51-L2 manifests mainly during and/or after the synaptic phase.

**dsDNA binding is severely impaired by mutations in the L1 loop or site II**. The binding of dsDNA by Rad51 is of critical importance during the homology search. Therefore, we analyzed the dsDNA binding activities of the mutants using fluorescence anisotropy (Fig. 6a and Supplementary Table 3). The $K_D$ of wild-type Rad51 for dsDNA was 2.79 μM, ~7.3-fold larger than the corresponding value for ssDNA. The $K_D$ value of the L2 mutant was slightly increased compared to wild-type Rad51, whereas the L1 and S2 mutants had such low dsDNA binding activities that we could not calculate $K_D$ values for them. Importantly, the L1 mutant showed a near-complete loss of dsDNA binding while the S2 mutant retained some potential for dsDNA binding. Interestingly, analysis of dsDNA binding via EMSA demonstrated that, in addition to the L1 and S2 mutants, the L2 mutant also exhibited a weaker mobility shift than wild-type Rad51 (Fig. 6b). The fact that the EMSA pattern of the L2 mutant was substantially different from wild-type Rad51, despite having a similar $K_D$ value, raises the possibility that the conformation of

dsDNA filaments formed by Rad51-L2 are somehow different from those formed by wild-type Rad51.

**Rad51-L2 does not fully extend dsDNA**. The dsDNA in the postsynaptic filament is about 1.5x longer than B-form DNA[26,44,45]. To examine whether Rad51-L2 formed elongated postsynaptic filaments, we analyzed Rad51-promoted dsDNA extension by a FRET-based assay using dsDNA that was double-labeled as before (Fig. 6c). The $E_{FRET}$ value decreased by 0.459 in the presence of wild-type Rad51, indicating that dsDNA is elongated upon Rad51 binding (Fig. 6c and Supplementary Table 6). This corresponds to 1.37× the length of B-form DNA. Swi5-Sfr1 had a marginal effect on the $E_{FRET}$ value (1.39x B-form DNA). The $E_{FRET}$ value decreased further in the presence of AMP-PNP (1.44× B-form DNA). The omission of ATP from the reaction largely impaired dsDNA elongation (Supplementary Fig. 5c). Therefore, much like ssDNA elongation, dsDNA elongation requires ATP binding but not hydrolysis. Importantly, the decrease in $E_{FRET}$ was significantly smaller (0.370) for the Rad51-L2 postsynaptic filament (Fig. 6c, Supplementary Fig. 5d and Supplementary Table 6). This value corresponds to 1.28× the length of B-form DNA. Like wild-type Rad51, Swi5-Sfr1 did not have a substantial effect on the $E_{FRET}$ value of Rad51-L2 (1.30× B-form DNA). The reduction in the $E_{FRET}$ value for Rad51-L2 was less than wild-type Rad51 even in the presence of AMP-PNP (1.33× B-form DNA compared to 1.44× for wild-type Rad51). These differences in the values between wild-type Rad51 and the L2 mutant are statistically significant. Therefore, these results clearly indicate that Rad51-L2 is impaired for elongation of

dsDNA, which likely explains why Rad51-L2 is defective in the C1–C2 transition.

## Discussion

It is difficult to envision how the Rad51 filament catalyzes the DNA strand exchange reaction. In this study, to decipher the molecular mechanisms underlying the DNA strand exchange reaction, we applied various FRET-based assay systems to analyze three DNA binding site mutants of Rad51: Rad51-L1, Rad51-L2, and Rad51-S2. Rad51-L1 and Rad51-L2 mutants have single mutations at conserved amino acid residues in L1 and L2 in site I, respectively, whereas the Rad51-S2 mutant has mutations at two conserved amino acid residues in site II, both of which have been proposed to play an equally important role in DNA binding[15,25]. The three mutant proteins exhibited distinct characteristics with respect to wild-type Rad51.

The Rad51-L1 (R257A) mutant displayed a near-complete loss of DNA strand exchange activity (Figs. 1d and 2). Although Rad51-L1 showed slightly decreased affinity for ssDNA (1.85-fold higher $K_D$ than wild-type Rad51) (Supplementary Table 3), the presynaptic filaments formed by Rad51-L1 showed comparable properties to those of wild-type Rad51, including the stretched ssDNA conformation and responsiveness to Swi5-Sfr1 (Fig. 5). However, this mutant showed a severe defect in binding to dsDNA (Fig. 6a, b) and was consequently unable to form C1. These findings suggest that Arg-257 is important for capture of dsDNA and formation of the C1 intermediate (Fig. 7a).

The Rad51-L2 (V295A) mutant formed C1 but the subsequent reaction was much slower than that of wild-type Rad51, even in the presence of Swi5-Sfr1 (Fig. 2 and Supplementary Table 2). The abortive strand exchange assay revealed that Rad51-L2 accumulated higher levels of C1 than wild-type Rad51 (Fig. 3). Consistently, the DNA strand exchange assay with 2AP directly

demonstrated that Rad51-L2 formed substantially lower levels of the C2 intermediate than wild-type Rad51 (Fig. 4). Presynaptic filaments formed by Rad51-L2 exhibited properties similar to those of the wild-type protein (Fig. 5) and Rad51-L2 retained dsDNA binding activity (Fig. 6a, b). Therefore, we conclude that the V295A mutation specifically impairs the C1–C2 transition (Fig. 7a). Notably, Rad51-L2 did not fully stretch ssDNA or dsDNA. However, while the defect in ssDNA extension could be overcome by the inclusion of Swi5-Sfr1 or AMP-PMP, the defect in dsDNA extension could not, pointing towards a critical role for V295 in dsDNA elongation (Figs. 5d and 6c). Both RecA-dsDNA crystallography data[15] and cryo-EM observations of human Rad51-dsDNA complexes[26] suggested that the corresponding amino acid side-chain inserts in-between two triplets to stabilize dsDNA. Taken together, we propose that V295 in the L2 loop promotes the strand exchange reaction by stabilizing hetero-duplex DNA in the C2 intermediate.

Like the L1 mutant, the Rad51-S2 (R324A and K334A in site II) mutant showed a near-complete loss of DNA strand exchange activity (Figs. 1d and 2). The mutant did not yield a calculable $K_D$ value for dsDNA (Fig. 6a and Supplementary Table 3) and the $k_1$ and $k_{-1}$ values in the strand exchange reaction containing Rad51-S2 were drastically reduced, indicating a severe defect in C1 formation (Fig. 2 and Supplementary Table 2). These observations are consistent with the notion that site II is involved in dsDNA binding, as proposed previously[14,16]. Interestingly, the mutant also exhibited a drastically reduced affinity for ssDNA (Fig. 5 a, b and Supplementary Fig. 3a), indicating that site II is important not only for dsDNA binding but also for ssDNA binding. Furthermore, the presynaptic filament formed by Rad51-S2 displayed an off-rate similar to wild-type Rad51, and the S2 filament was elongated similarly to the wild-type filament. In addition, Swi5-Sfr1-mediated stabilization was normal for the presynaptic filament formed by Rad51-S2 (Supplementary

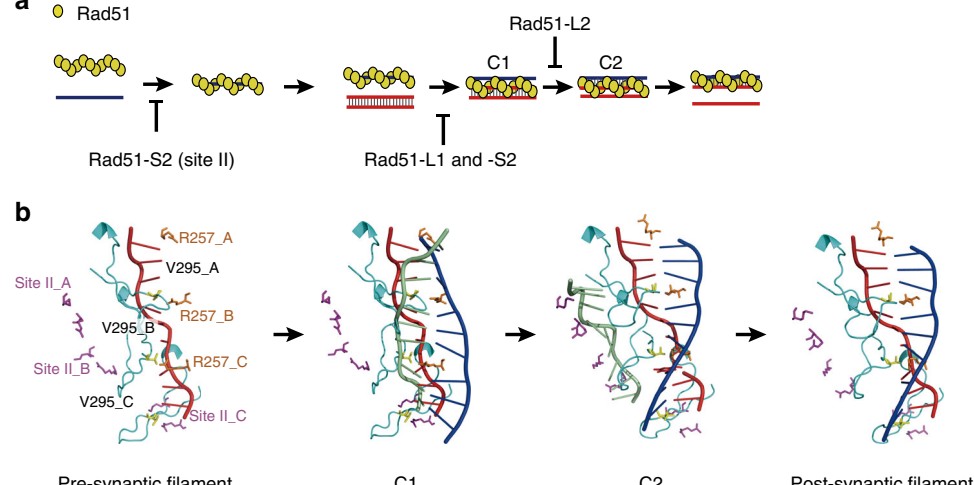

**Fig. 7 A model of DNA strand exchange reaction driven by Rad51. a** Roles of each DNA binding site of Rad51 in the DNA strand exchange reaction. Yellow circles, black lines, and red double-lines represent Rad51 monomers, ssDNA, and donor dsDNA molecules, respectively. **b** A model of the strand exchange reaction proceeding in order of the presynaptic filament (three Rad51 monomers binding to ssDNA are shown), C1, C2, and the post-synaptic filament. Only Arg-257 (orange) in L1, Val-295 (yellow) in L2 (light blue), and site II (Arg-324 and Lys-334, purple) are shown. A structural model of C1 was constructed by docking the SpRad51-ssDNA filament model shown in Fig. 1b and extended-dsDNA. The incoming dsDNA retains its duplex base-pairs and the invading ssDNA (red) aligns (but does not pair) with the complementary strand (blue) of the incoming dsDNA. Arg-257 in L1 and Val-295 in L2 of the presynaptic filament is arranged in close proximity with the complementary strand and the invading strand, respectively. Because of steric hindrance observed in the initial model (Supplementary Fig. 6a), the dsDNA is separated from site II (Arg-324 and Lys-334). A structural model of C2 was constructed by docking the SpRad51-dsDNA filament model shown in Fig. 1c and extended-ssDNA. A heteroduplex composed of the invading ssDNA (red) and the complementary strand (blue) of the incoming dsDNA is already engaged in base-pairing and the non-complementary ssDNA (green) is displaced from the incoming dsDNA. Arg-257 in L1 and Val-295 in L2 insert into the incoming dsDNA from opposite directions, stabilizing the heteroduplex DNA. Site II interacts with the displaced ssDNA. Source data are provided as a Source data file.

Table 4). Thus, the observed low affinity was because the on-rate of Rad51-S2 onto ssDNA was much lower than that of the wild-type protein (Fig. 5). Taken together, these observations suggest that the deficiency in strand exchange caused by this mutation is primarily due to a defect in binding to both ssDNA and dsDNA, and not to a defect in the catalytic function required for the formation of heteroduplex DNA (Fig. 7a). Therefore, we posit that site II serves as an entry gate for both ssDNA and dsDNA.

To explore the mechanisms involved, we constructed structural models of the two reaction intermediates using docking simulations. In the C1 complex, the donor dsDNA and the ssDNA in the presynaptic filament align without engaging in base-pairing. We therefore tried to make a structural model of C1 by combining the SpRad51-ssDNA filament modeled in Fig. 1b and extended B-form DNA with minimal steric hindrances. However, when Arg-257 in L1 was inserted into the inter-triplet gap of dsDNA, as suggested by Xu et al. [26], and site II was oriented to interact with dsDNA, a collision between L2 and dsDNA occurred (Supplementary Fig. 6). The only way to avoid this collision was to pass one strand of the dsDNA through the L2 loop or to move the dsDNA away from L2. Since the former configuration is not possible for C1, we adopted the latter option. As a result, dsDNA was separated from site II. The resultant C1 model is shown in Fig. 7b. In this model, the side chain of Val-295 inserts in-between two triplets of elongated ssDNA in the original presynaptic complex, as reported previously [15,26].

In the C2 complex, the initial invading ssDNA in the presynaptic filament is intertwined with the complementary strand of the donor dsDNA. In line with this configuration, a docking simulation was performed between the extended ssDNA corresponding to the displaced strand and the post-synaptic filament modeled in Fig. 1b with minimal steric hindrances (Fig. 7b). This resulted in the interaction of site II with the displaced strand without any significant steric hindrance. In the resultant C2 structure, Arg-257 in L1 and Val-295 in L2 insert into the dsDNA from opposite directions, stabilizing the heteroduplex DNA, as previously suggested [26]. These observations imply that Arg-257 and Val-295 may help to recruit the complementary strand and lower the energy state of the complementary strand for base-pairing during the homology search [26].

Based on the results described above and the available structural information, we propose the following model for the strand exchange reaction. First, Arg-324 and Lys-334 in site II promote the capture of ssDNA to form the presynaptic filament. Subsequently, Arg-257 in L1 in the presynaptic filament mediates dsDNA capture, leading to C1 formation. The basic patch in site II also helps to act as an entry gate for dsDNA capture. Once Arg-257 in L1 inserts into dsDNA, the dsDNA separates from site II. Because strong static interactions between site II and dsDNA are disadvantageous for strand exchange, release of dsDNA from site II in the C1 complex must be important for progression to the C2 complex. After the release of dsDNA from site II, Val-295 in the L2 loop inserts into the incoming dsDNA to form inter-triplet gaps in dsDNA, which reduces stacking interactions. This likely leads to an increased potential for triplet flapping, which is utilized in the homology search. When the captured dsDNA has homology to the ssDNA, a de novo heteroduplex of 3 bp is transiently formed. If homology is extensive, sequential insertions of Arg-257 and Val-295 at each triplet results in heteroduplex extension, leading to C2 complex formation. This reaction corresponds to the C1–C2 transition. Importantly, Val-295 in the L2 loop is directly involved in this step, as shown by the results of the real-time assay (Fig. 2). In addition, Val-295 also stabilizes the resultant heteroduplex in the C2 complex. Site II also stabilizes the displaced ssDNA in the C2 complex, which functions in a complementary manner to prevent the reverse reaction from occurring.

Site II of RecA is thought to play critical roles in dsDNA capture and the homology search. It was previously thought that when the presynaptic filament encounters a homologous sequence, the complementary strand of the duplex DNA moves into site I to form a heteroduplex with the initial ssDNA; this process was thought to be aided by site II [25]. However, recent studies show that site II of RecA and HsRAD51 binds ssDNA more strongly than dsDNA and captures only one strand of dsDNA, leaving the unbound complementary strand available for base sampling [18,19,48]. In addition, the *S. cerevisiae* Rad51-II3A (ScRad51-II3A) mutant, which has two mutations within site II (K361A and K371A) and one in the Walker A motif (R188A), was reported to show reduced binding affinity for both ssDNA and dsDNA, as judged by EMSA [49]. In that sense, the ScRad51-Il3A mutant is similar to the SpRad51-S2 mutant presented here, although the former has an extra mutation within the Walker A motif. Moreover, Cloud et al. showed that the ScRad51-II3A presynaptic filament cannot bind to heterologous dsDNA [49], which is in good agreement with our demonstration of a severe defect in C1 formation for SpRad51-S2 (Fig. 2). Collectively, our results obtained using Rad51-S2 are essentially consistent with these recent observations. Because C2 complex formation requires Val-295 in the L2 loop but no residues in site II, the fundamental role of site II must be to provide a basic patch favorable for ssDNA and dsDNA binding.

Ile-199 of RecA and Val-273 of HsRAD51, both of which correspond to Val-295 in L2 of SpRad51, were shown to insert into the inter-triplet gap of both ssDNA and dsDNA filaments [15,26]. Our data clearly indicate that Val-295 is indeed important for elongation of both ssDNA and dsDNA. The cryo-EM structures of the HsRad51 filament indicates that Val-273 inserts more shallowly into the inter-triplet gap of ssDNA than that of dsDNA [26]. We expect that the shallow insertion of Val-295 is sufficient to stabilize the elongated ssDNA filament but that stabilization of the elongated dsDNA filament requires a deeper insertion of Val-295. Notably, the defect of the L2 mutant in ssDNA elongation could be completely overcome by Swi5-Sfr1 or AMP-PNP (Fig. 5d), both of which induce an active presynaptic filament [29,42]. In contrast, the impairment of the L2 mutant in dsDNA elongation was not suppressed by Swi5-Sfr1 or AMP-PNP. Since the L2 mutant has a severe defect in the C1–C2 transition, the observed impairment in dsDNA elongation implies that dsDNA elongation is critically important for the C1–C2 transition. Interestingly, both ssDNA and dsDNA elongations do not require ATP hydrolysis because the elongation occurs in the presence of AMP-PNP, while the C1–C2 transition requires ATP hydrolysis [27]. ATP hydrolysis may be associated with the movement of L2, which is originally located with the ssDNA in the presynaptic filament and then with dsDNA in the de novo heteroduplex.

The Rad51-L2 mutant showed reduced ATPase activity in the absence of DNA or in the presence of ssDNA. However, it did not show a defect in ATP hydrolysis in the presence of dsDNA or in the presence of both ssDNA and dsDNA (Fig. 1f). The ATP hydrolysis defect of the L2 mutant cannot be explained by an inability to bind ssDNA since the L2 mutant bound ssDNA similarly to wild-type Rad51 (Fig. 5). More importantly, wild-type Rad51 of *S. pombe* exhibits substantial ATPase activity even in the absence of ssDNA (Fig. 1f), as reported previously [29,42,50], unlike ScRad51 or HsRad51 [51,52]. Therefore, we surmise that the low ATPase activity of L2 in the presence of ssDNA is not the primary reason for the inability of the L2 mutant to drive the C1–C2 transition. While the ATPase activity of wild-type protein was marginally stimulated by ssDNA, this ATPase stimulation was not observed for the L2 mutant. Insertion in-between ssDNA intertriplets by Val-295, which is expected not to occur in the L2

**Table 1 Oligo DNA substrates used in this study.**

| Name | Sequence (5′ to 3′) |
|------|---------------------|
| 16A(−) | AAATGAACATAAAGTAAATAAGTATAAGGATAATACAAAATAAGTAAATGAATAAACATAGAAAATAAAG TAAA GGATAT AAA |
| [FAM]16A(−) | [FAM]AAATGAACATAAAGTAAATAAGTATAAGGATAATACAAAATAAGTAAATGAATAAACATAGAAAATAAAG TAAAGGATAT AAA |
| 16A(−)_3 × 2AP | AAATG[2AP]ACATAAAGTAAAT[2AP]AGTATAAGGATAAT[2AP] CAAAATAAGTAAATGAATAAACATAGAAAATAAAGTAAAGGATATAAA |
| 16A(−)_40 bp | AAATGAACATAAAGTAAATAAGTATAAGGATAATACAAAA |
| [FAM]16A(−)_40 bp | [FAM]AAATGAACATAAAGTAAATAAGTATAAGGATAATACAAAA |
| 16A(+)_40bp | TTTTGTATTA TCCTTATACT TATTTACTTT ATGTTCATTT |
| 16A(+)[ROX]_40 bp | TTTTGTATTA TCCTTATACT TATTTACTTT ATGTTCATTT-[ROX] |
| [TAMRA]dT72 | [TAMRA]TTTTTTTTTTTTTTTTTTTTTTTTTTTTTTTTTTTTTTTTTTTTTTTTTTTTTTTTTTTTTTTTTTTTTTTTTTTT |
| [TAMRA]16A(−)_72 bp | [TAMRA]AAATGAACATAAAGTAAATA AGTATAAGGATAATACAAAA TAAGTAAATGAATAAACATAGAAAATA AAGTA |
| 16A(+)_72 bp | TACTTTATTTTCTATGTTTATTCATTTACTTATTTTGTAT TATCCTTATACTTATTTACTTTATGTTCATTT |
| 16(A)(−)_F + R_73 bp | AAATGAACATAAAGTAAATAAGTATAAGGA[Fluorosein-dT]AATAC AAAAT[ROX-dT] AAGTAAATGAATAAACATAGAAAATAAAGTA |
| 16(A)(+)_73bp | TACTTTATTTTCTATGTTTATTCATTTACTTAATTTTGTATTATCCTTATACTTATTTACTTTATGTTCATTT |

mutant, may induce the stimulation of the ATPase by ssDNA, thus potentially explaining why the L2 mutant shows a loss of ssDNA-stimulated ATP hydrolysis.

Recent studies have suggested that Rad51 filaments may function to protect DNA at replication forks by impeding nucleolytic digestion[53,54]. Importantly, the ability of Rad51 to drive DNA strand exchange appears to be dispensable for this role. Since the L1 and L2 mutants were mostly proficient for ssDNA binding and capable of forming stable presynaptic filaments (Fig. 5a, c), it would be intriguing to examine whether they are proficient for replication fork protection.

In conclusion, because the three steps of the strand exchange reaction can now be analyzed using multiple real-time assay systems, our results provide new insights into the reaction mechanism, which could not be dissected with conventional assays. The conceptual models presented here provide a framework for a detailed understanding of the precise molecular mechanisms underlying Rad51-dependent homologous recombination.

## Methods

**Proteins**. Rad51, Swi5-Sfr1 complex and RPA complex (Rpa1–Rpa2–Rpa3) were over-expressed using the *E. coli* strain BL21-Codonplus(DE3) RIPL (Agilent). Expression conditions were the same in each case (induced with 1 mM isopropyl β-D-1-thiogalactopyranoside [IPTG] for ~15 h at 18 °C). Cell pellets were resuspended in R buffer (20 mM Tris-HCl pH 8.0, 1 mM EDTA, 1 mM DTT and 10 % glycerol) containing 300 mM NaCl and disrupted by sonication. The whole cell extract was centrifuged at $50,000 \times g$ for 1 h. Subsequent steps varied depending on the protein, as described below.

For Rad51, ammonium sulfate was added to the supernatant to 35% saturation and the mixture was centrifuged at $18,000 \times g$ for 1 h. The pellet was then resuspended in P buffer (20 mM potassium-phosphate pH 7.5, 0.5 mM EDTA, 0.5 mM DTT and 10% glycerol) containing 100 mM KCl and the mixture was subjected to an SP Sepharose (GE healthcare) column. The flow-through fraction was collected and loaded onto a Q Sepharose (GE healthcare) column and proteins were eluted with a liner gradient from 100 to 800 mM KCl. Peak fractions of Rad51 protein appeared at ~400 mM KCl. The fractions were diluted 4-fold with P buffer and loaded onto a Hitrap Heparin column (GE Healthcare). Rad51 eluted at ~300 mM KCl. Peak fractions were diluted 3-fold with P buffer and loaded onto a Resouce Q (GE Healthcare) column. Rad51 eluted at about 400 mM KCl. Peak fractions were combined, dialyzed against P buffer containing 200 mM KCl, concentrated using an Amicon Ultra-4 (10k) centrifugal filter, and small aliquots were frozen in liquid nitrogen before storing at −80 °C. Protein concentration was determined by measuring the absorbance at 280 nm (ε = 18,600).

The Rad51-L2 and Rad51-S2 mutant proteins were purified using the same procedure used for wild-type Rad51. For Rad51-L1, a HiTrap Blue column was used instead of a HiTrap Heparin column (GE Healthcare) after Q sepharose chromatography. Proteins were eluted with a linear gradient from 0 M to 2 M KCl

in P buffer. Other purification steps of the Rad51-L1 protein were the same as wild-type Rad51.

For Swi5-Sfr1, ammonium sulfate was added to the supernatant to 40% saturation and the mixture was centrifuged at $18,000 \times g$ for 1 h. The pellet was resuspended in R buffer containing 100 mM NaCl and the mixture was subjected to a Q Sepharose (GE Healthcare) column and proteins were eluted with a liner gradient from 100 to 1000 mM NaCl. Peak fractions of the Swi5-Sfr1 complex appeared at ~300 mM NaCl. The fractions were diluted 3-fold with R buffer and loaded onto a Hitrap Heparin (GE Healthcare) column. Swi5-Sfr1 complex eluted at ~600 mM NaCl. The eluted fractions were loaded onto a 16/60 Superdex 200 pg gel filtration column (GE Healthcare) and developed in R buffer containing 1 M NaCl. Peak fractions were combined, dialyzed against R buffer containing 200 mM NaCl, concentrated using an Amicon Ultra-4 (10 k) centrifugal filter, and small aliquots were frozen in liquid nitrogen before storing at −80 °C. Protein concentration was determined by measuring the absorbance at 280 nm (ε = 12,615).

For RPA, ammonium sulfate was added to the supernatant to 45% saturation and the mixture was centrifuged at $18,000 \times g$ for 1 hr. The pellet was resuspended in R buffer containing 50 mM NaCl and the mixture was subjected to an SP Sepharose (GE Healthcare) column and proteins were eluted with a liner gradient from 50 to 1000 mM NaCl. Peak fractions of RPA appeared at ~300 mM NaCl. These fractions were diluted 3-fold with R buffer and loaded onto a Hitrap Heparin (GE Healthcare) column. RPA eluted at ~600 mM NaCl. The eluted fractions were loaded onto a 16/60 Superdex 200 pg gel filtration column (GE Healthcare) and developed in R buffer containing 1.5 M NaCl. Peak fractions were combined, dialyzed against R buffer containing 100 mM NaCl, concentrated using an Amicon Ultra-4 (10k) centrifugal filter, and small aliquots were frozen in liquid nitrogen before storing at −80 °C. Protein concentration was determined by measuring the absorbance at 280 nm (ε = 97,640).

**DNA substrates**. The oligonucleotides used in this study are listed in Table 1. The 83-mer oligonucleotide 16A(−) was used to form the presynaptic filament in the pairing and displacement assays[27,31,55]. To prepare donor dsDNA for the pairing and displacement assays, 16A(−) was truncated 43 bases from the 3′ end to yield 16A(−)_40bp, which was annealed with its complementary strand. Real-time analysis of the DNA strand exchange reaction using 2-aminopurine (2AP) was performed using 16A(−)_3 × 2AP. The ssDNA binding assay was performed using 5′ TAMRA-labeled oligo dT (72-mer). The dsDNA binding assay was performed with 5′ TAMRA-labeled 16A(−)_72 bp, which was truncated 11 bases from the 3′ end and annealed with its complementary strand. To investigate the conformation of Rad51-ssDNA and -dsDNA filaments by FRET, 16(A)(−)_F + R_73bp was labeled internally with fluorescein and rhodamine (annealed to its unlabeled complementary strand in the case of dsDNA). All oligo DNAs were purchased from Eurofins Genomics.

**Real-time DNA strand pairing and displacement assays**. All reactions were carried out in buffer A (30 mM HEPES-KOH [pH 7.5], 15 mM MgCl$_2$, 1 mM DTT, 0.1 % [w/v] BSA, 0.0075 % Tween-20, 0.25 mM ATP). In the pairing assay, Rad51 protein (1.5 μM) was added to buffer A (1.6 ml) containing fluorescein-labeled ssDNA (36 nM, fragment concentration) at 37 °C. After incubation for 5 min, Swi5-Sfr1 (0.15 μM) was added to the reaction mixture and incubation was continued for another 5 min. The mixture (1.5 ml) was transferred to a cuvette set in a

spectrofluorometer (FP-8300, JASCO). The pairing reaction was started by injecting rhodamine-labeled dsDNA (36 nM fragment concentration) into the reaction mixture. The change in fluorescence of fluorescein was monitored at 525 nm upon excitation at 493 nm. In the displacement assay, the reaction procedure was the same as in the pairing assay, except that the reaction volume was smaller (130 µl) and fluorescein- and rhodamine-labeled dsDNA was added manually. In the case of pairing assays, the experimental data were normalized using the equation described below, where $F_{raw}$ is the fluorescence intensity from raw data and $F_{change\_p}$ is the change in fluorescence, calculated using Eq. (1).

$$(F_{change\_p}) = (F_{raw} \text{ at time } t) \times (F_{raw} \text{ at time } 0)^{-1} \quad (1)$$

$F_{raw}$ at time 0 is the average fluorescence monitored for the last 20 s before the reaction was initiated by addition of the dsDNA substrate.

Substrate % was calculated using Eq. (2).

$$(\text{Substrate \%}) = \left\{ 1 - [(F_{change\_p} \text{ without protein}) - (F_{change\_p} \text{ with Rad51})] \right. \\ \left. \times (1 - E_{max})^{-1} \right\} \times 100 \quad (2)$$

Maximum FRET efficiency ($E_{max}$) of the pairing assay was calculated to be 0.317.

$E_{max}$ of the pairing assay was calculated using Eq. (3).

$$(E_{max}) = (\text{emission at 525 nm dsDNA double} - \text{labeled by fluorescein and ROX with Rad51}) \\ \times (\text{emission at 525 nm ssDNA labeled by fluorescein with Rad51})^{-1} \quad (3)$$

In the case of displacement assays, the experimental data were normalized using the equation described below, where $F_{raw}$ is the fluorescence intensity from raw data and $F_{change\_d}$ is the change in fluorescence, calculated using Eq. (4).

$$(F_{change\_d}) = [(F_{raw} \text{ at time } t) - (F_{raw} \text{ at time } 0)] \\ \times [(F_{raw} \text{ at time } 0) \times E_{max}^{-1} - (F_{raw} \text{ at time } 0)]^{-1} \quad (4)$$

$F_{raw}$ at time 0 was the average emission measured for the first 5 s after the dead time.

Product % was calculated using Eq. (5).

$$(\text{Product \%}) = [(F_{change\_d} \text{ of sample}) - (F_{change\_d} \text{ of control without Rad51})] \times 100 \quad (5)$$

$E_{max}$ of the displacement assay was calculated to be 0.279. The equation for calculating $E_{max}$ of the displacement assay is the same that used for the pairing assay.

**Abortive DNA strand exchange assay**. The reaction procedure was the same as for the pairing assay, except that the reaction volume was smaller (150 µl) and homologous dsDNA was added manually. Five, 10, or 20 min after the reaction started, EDTA (50 mM) was added to dissociate three-strand intermediates. The reaction was monitored as described above. Substrate % was calculated by the same method as the pairing assay. Amount of C1 % was calculated using Eq. (6), where $S_{before}$ is the amounts of substrates before addition of EDTA and $S_{after}$ is the amounts of substrates 200 s after addition of EDTA.

$$(\text{Amount of C1 \%}) = 100 \times (S_{after} - S_{before}) \times (100 - S_{before})^{-1} \quad (6)$$

**Real-time DNA strand pairing using DNA containing 2AP**. The procedure for real-time analysis of DNA strand pairing using DNA containing 2AP was essentially the same as that for the DNA strand displacement assay, except that the presynaptic filament was formed on 16A(−)_3 × 2AP and non-labeled dsDNA was added to initiate the reaction. The change in fluorescence of 2AP was monitored at 375 nm upon excitation at 310 nm. Data were collected every second. The experimental data were normalized using the equation described below, where $F_{raw}$ is the fluorescence intensity from raw data and $F_{change\_2AP}$ is the change in fluorescence, calculated using Eq. (7).

$$(F_{change\_2AP}) = (F_{raw} \text{ at time } t) \times (F_{raw} \text{ at time } 0)^{-1} \quad (7)$$

$F_{raw}$ at time 0 is the average fluorescence monitored for the last 20 s before the reaction was initiated by addition of the dsDNA substrate.

Substrate % was calculated using Eq. (8).

$$(\text{Substrate \%}) = \left\{ 1 - [(F_{change\_2AP} \text{ without protein}) \right. \\ \left. - (F_{change\_2AP} \text{ with Rad51})] \times (1 - Q_{max})^{-1} \right\} \times 100 \quad (8)$$

Product % was calculated using Eq. (9)

$$(\text{Product \%}) = 100 - (\text{average of substrate \% from 1496 to 1500 s}) \quad (9)$$

$Q_{max}$ is the maximum quenching efficiency of 2AP in the pairing assay, which was calculated using Eq. (10).

$$(Q_{max}) = (\text{emission of 2 AP labeled dsDNA containing 2 AP} - \text{T base pairs}) \\ \times (\text{emission of 2 AP labeled ssDNA})^{-1} \quad (10)$$

The $Q_{max}$ value did not differ significantly between wild-type Rad51 and Rad51-L2 (Supplementary Table 7).

**DNA binding assay using fluorescence anisotropy**. In the ssDNA binding assay, the fluorescence anisotropy of the reaction solution containing 3 µM (nucleotide concentration) TAMRA-labeled oligo dT, and 1 mM ATP in buffer B (30 mM HEPES-KOH [pH 7.5], 100 mM KCl, 3 mM MgCl₂, 1 mM DTT and 5% glycerol) was measured at 575 nm upon excitation at 546 nm in an FP-8300 spectrofluorometer (JASCO). Rad51 was added to the reaction solution and incubated for 5 min at 25 °C. The fluorescence anisotropy of the reaction was then measured, more Rad51 was added to the reaction solution, and the same manipulation described above was repeated. The Rad51 concentration was titrated as indicated in Fig. 5a. For the dsDNA binding assay, the experimental procedure was the same as for the ssDNA binding assay, except that the reaction solution contained 3 µM (bp) TAMRA-labeled dsDNA and a different titration series of Rad51 was used, as indicated in Fig. 6a. Based on the experimental data, a dissociation constant was calculated using Eq. (11), in which [Rad51] is the concentration of Rad51 and $n$ is the Hill coefficient.

$$(\text{Anisotropy}) = (\text{Minimum value of fluorescence anisotropy}) \\ + ([\text{Amplitude of change in fluorescence anisotropy}] \times [\text{Rad51}]^n) \\ \times ([\text{Rad51}]^n + K_D^n)^{-1} \quad (11)$$

**Fluorescence anisotropy for Rad51 binding kinetics**. In the standard association assay, a $1.0 \times 1.0$ cm cuvette containing 2 ml reaction buffer B plus 1 mM ATP and 1.2 µM (nucleotide) TAMRA-labeled oligo dT was set in the spectrofluorometer at 25 °C. The reaction mixture was stirred continuously at 450 rpm with a magnet stirrer. The fluorescence anisotropy of TAMRA was monitored once per second at 575 nm upon excitation at 546 nm. Sixty seconds after the measurement started, Rad51 was injected into the reaction mixture at a final concentration of 0.4 µM. After incubation for 5 min, the indicated concentration of Swi5-Sfr1 was injected into the reaction and the incubation continued for another 5 min.

In the dissociation assay, a $1.0 \times 1.0$ cm cuvette containing 2 ml reaction buffer B plus 1 mM ATP, without protein or DNA, was set in the spectrofluorometer at 25 °C. The reaction mixture was stirred continuously at 450 rpm. with a magnet stirrer. Sixty seconds after the measurement started, 50 µl of reaction mixture from the association assay was injected into the cuvette. Fluorescence anisotropy (FA) was monitored for 1000 s every 1 s. Based on the results of the dissociation assay, a dissociation rate constant ($k_{off}$) was calculated using Eq. (12), using KaleidaGraph, ver 4.5.1 (Hulinks).

$$(FA[t]) = (\text{Amplitude of change in FA}) \times EXP(-k_{off} \times t) + (\text{Minimum value of FA}). \quad (12)$$

FA($t$) indicates FA at time t (s).

**Extension of DNA upon Rad51 binding**. ssDNA internally double-labeled with fluorescein and ROX (3 µM nucleotide concentration) was mixed with reaction buffer B plus 1 mM ATP or 1 mM AMP-PNP, and the mixture was incubated at 25 °C. After 5 min, the emission spectrum was measured from 500 to 650 nm upon excitation at 493 nm. After the first measurement, 3 µM Rad51 was added to the reaction mixture and the solution was incubated at 25°C. After 5 min, the emission spectrum was measured under the same conditions described above. After the second measurement, 0.3 µM Swi5-Sfr1 was added to the reaction solution. After 5 min, the emission spectrum was measured under the same conditions described above. Normalized Int. was calculated using Eq. (13).

$$(\text{Normalized Int.}) = (\text{fluorescence intensity at} \times \text{nm of each condition}) \\ \times (\text{fluorescence intensity at 525 nm of DNA only})^{-1} \quad (13)$$

FRET efficiency ($E_{FRET}$) was calculated using Eq. (14). Constants used to calculate FRET efficiency are shown in Supplementary Table 8. $I_D$ is the fluorescence emission of donor (fluorescein) at 525 nm; $I_A$ is the fluorescence emission of acceptor (ROX) at 605 nm, $\varphi_D$ and $\varphi_A$ are the quantum yields of donor and acceptor, respectively; and $I_{605/525}$ indicates how the emission of the donor affects the emission at 605 nm.

$$(E_{FRET}) = I_A \times \{ I_A + (\phi_A \times \phi_D^{-1}) \times I_D \}^{-1} \quad (14)$$

$$I_D = (\text{fluorescence intensity at 525 nm})$$

$$I_A = \{ (\text{fluorescence intensity at 605 nm}) - (I_{605/525} \times \text{fluorescence intensity at 525 nm}) \} \quad (15)$$

To obtain the values of $\phi_A$ per $\phi_D$ and $I_{605/525}$, fluorescein or ROX single-labeled ssDNA was prepared and subjected to the experiments described below. Fluorescein-labeled ssDNA (3 μM nucleotide concentration) was mixed with reaction buffer B plus 1 mM ATP and the mixture was incubated at 25 °C. After 5 min, the emission spectrum was measured from 460 to 700 nm upon excitation at 450 nm. After the first measurement, 3 μM Rad51 was added to the reaction mixture, and the solution was incubated at 25 °C. After 5 min, the emission spectrum was measured under the same conditions described above. After the second measurement, 0.3 μM Swi5-Sfr1 was added. After 5 min, the emission spectrum was measured under the same conditions described above. In the case of ROX-labeled ssDNA, the experimental procedure was the same as for fluorescein-labeled ssDNA, except that the emission spectrum of ROX was measured from 535 to 750 nm upon excitation at 530 nm. The absorbance of fluorescein-labeled ssDNA at 450 nm, and that of ROX-labeled ssDNA at 530 nm, was measured using NanoDrop 2000c (Thermo Fisher). The values of $\phi_A$ per $\phi_D$ were calculated using Eq. (16), where $A_D$ is the area of the emission spectra of fluorescein-labeled ssDNA; $A_A$ is the area of the emission spectra of ROX-labeled ssDNA; $B_D$ is the absorbance of fluorescein-labeled ssDNA at 450 nm; $B_A$ is the absorbance of ROX-labeled ssDNA at 530 nm; $S_{450}$ is the intensity of excitation light at 450 nm; and $S_{530}$ is the intensity of excitation at 530 nm.

$$\phi_A \times \phi_D^{-1} = \left(A_A \times A_D^{-1}\right) \times \left(S_{450} \times S_{530}^{-1}\right) \times \left(B_D \times B_A\right)^{-1}, S_{450} \times S_{530}^{-1}$$
$$= 1.66 \text{ and } B_D \times B_A^{-1} = 1.52 \quad (16)$$

$I_{605/525}$ was calculated using Eq. (17).

$$(I_{605/525}) = (\text{emission of fluorescein} - \text{labeled ssDNA at 605 nm upon excitation at 450 nm})$$
$$\times (\text{the emission of fluorescein} - \text{labeled ssDNA at 525 nm upon excitation at 450 nm})^{-1} \quad (17)$$

For analysis of the Rad51-dsDNA filament, the experimental procedures were the same as for the Rad51-ssDNA filament except for the concentrations of dsDNA (3 μM bp concentration), Rad51 (8 μM), and Swi5-Sfr1 (0.8 μM). Constants used to calculate FRET efficiency are shown in Supplementary Table 9. For the Rad51-dsDNA filament assay, the value of $B_D$ per $B_A$ was 1.32.

To estimate the distance between fluorescein and ROX on DNA, the Förster distance ($R_o$) of two fluorophores on ssDNA or dsDNA was calculated using Eq. (18).

$$(R_o) = 0.211 \times \left[\left(\kappa^2 \times J(\lambda) \times \phi_F\right) \times n^{-4}\right]^{1/6} \quad (18)$$

$\kappa^2$ is the dipole orientation factor and we used $\kappa^2 = 2/3$ because donor and acceptor randomly rotate in solution. $\varphi_F$ is quantum yield of fluorescein and we used $\varphi_F = 0.85$. $n$ is the refractive index of the water medium, we used $n = 1.57$. $J(\lambda)$ is the overlap integral of the fluorescence emission spectrum of fluorescein and the absorption spectrum of ROX. The $J(\lambda)$ and $R_o$ values determined are shown in Supplementary Table 10. The molar absorption coefficient for ROX at 585 nm ($\varepsilon_{585}$) for calculation of $J(\lambda)$ is 78000.

The distance between the two fluorophores on DNA was estimated using Eq. (19), in which $r$ is the distance between two fluorophores.

$$(E_{FRET}) = R_o^6 \times \left(R_o^6 + r^6\right)^{-1} \quad (19)$$

Using Supplementary Tables 6 and 10 and Eq. (19), the distance of two fluorophores in B-form dsDNA without protein was estimated (41.2 Å). We calculated the values of orange lines in Figs. 5d and 6c using this distance.

**Conventional three-strand exchange assay using long DNAs.** First, 5 μM Rad51 and 10 μM (nucleotide) φX174 circular ssDNA (NEB) were mixed in reaction buffer C (30 mM Tris-HCl [pH 7.5], 100 mM KCl, 3 mM MgCl$_2$, 1 mM DTT, 5% glycerol) containing 2 mM ATP and an ATP regeneration system (8 mM creatine phosphate and 8 U/ml creatine kinase), and the mixture was incubated at 37 °C. After 5 min, the indicated concentration of Swi5-Sfr1 was added, and incubation was continued for 5 min, followed by addition of 1 μM RPA. After 10 min, 10 μM (nucleotide) linear φX174 dsDNA (NEB) was added to the mixture to initiate the three-strand exchange reaction. After 120 min, 200 μg of psoralen was added to the reaction and the reaction mixtures were exposed to UV to cross-link DNA. After DNA cross-linking, 1.8 μl of reaction stop solution containing 5.3% SDS and 6.6 mg/ml proteinase K (Takara) was added and the sample was incubated for 60 min at 37 °C. Substrates and reaction products were separated by agarose gel electrophoresis and the gel was stained with SYBR-Gold.

**Electrophoresis mobility shift assay (EMSA).** Various concentrations of Rad51 as indicated and 10 μM (bp) linearized φX174 dsDNA or φX174 circular ssDNA were mixed in reaction buffer B plus 2 mM ATP and the mixture incubated at 37 °C. After 15 min, glutaraldehyde (0.2% [w/w]) was added and incubation was continued for 5 min. Free DNA and DNA in complex with Rad51 were separated by 1 % agarose electrophoresis and the gel was stained with SYBR-Gold.

**Rad51 ATP binding assay.** The ATP binding assay used TNP-ATP, an ATP analog whose fluorescence emission increases when it interacts with a protein and is incorporated into a hydrophobic environment[56]. In a 0.2 × 1.0 cm cuvette, TNP-ATP (Tocris Bioscience) at a final concentration of 0.5 μM was mixed with 120 μl

of reaction buffer B. The cuvette was set in the spectrofluorometer and incubated at 4 °C. After 1 min, the fluorescence spectrum of TNP-ATP was measured from 500 to 600 nm upon excitation at 485 nm. After the baseline fluorescence was measured, various concentrations of Rad51 were added and the mixture was incubated at 4 °C. After 5 min, the fluorescence spectrum of TNP-ATP was measured from 500 to 600 nm upon excitation at 485 nm. The normalized fluorescence was calculated using Eq. (20).

$$(\text{Normalized fluorescence}) = [(\text{fluorescence intensity of TNP} - \text{ATP at 540 nm with Rad51})$$
$$\times (\text{fluorescence intensity of TNP} - \text{ATP at 540 nm without Rad51})^{-1}] - 1 \quad (20)$$

The values of $K_D$ were calculated using Eq. (21), using KaleidaGraph, ver 4.5.1 (Hulinks). $\Delta F$ is the amplitude of change in the normalized fluorescence, [TNP-ATP] is the concentration of TNP-ATP, and [Rad51] is the concentration of Rad51.

$$(\text{Normalized fluorescence}) = \Delta F \times \{(K_D + [\text{TNP} - \text{ATP}] + [\text{Rad51}])$$
$$- [(K_D + [\text{TNP} - \text{ATP}] + [\text{Rad51}])^2 - 4[\text{TNP} - \text{ATP}] \times [\text{Rad51}]]^{1/2}\} \times (2[\text{TNP} - \text{ATP}])^{-1} \quad (21)$$

**Analysis of ATP hydrolysis of Rad51.** Reaction mixtures (80 μl) containing 5 μM Rad51, 0.5 μM Swi5-Sfr1, and 10 μM (nucleotide) φX174 DNA (ssDNA, dsDNA or both) in buffer C were prepared on ice. The reactions were started by addition of ATP at a final concentration of 0.5 mM and incubated at 37 °C. Aliquots (10 μl) were taken at various time points (0, 5, 10, 15, 20, 30, and 40 min) and mixed with 2 μl of 120 mM EDTA to stop the reaction. Inorganic phosphate generated by ATP hydrolysis was detected using the Malachite Green Phosphate Assay Kit (BioAssay Systems).

**Structural models of C1 and C2 complexes.** For the C1 complex, the presynaptic filament of SpRad51 was constructed by aligning the core domains of SpRad51 with those of the cryo-EM structure of the HsRAD51-ssDNA filament (PDB ID: 5H1B)[26] as a reference, using MODELLER software ver. 9.17[57]. This structure is shown in Fig. 1b. Then, the 1.5-fold-extended dsDNA was docked manually to the modeled SpRad51-ssDNA filament. Arg-257 in the complex was then inserted into the inter-triplet gap of dsDNA by MODELLER. Because of the steric hindrance observed in the initial model (Supplementary Fig. 6a), the dsDNA is manually separated from site II. To construct the C2 complex, the postsynaptic filament of SpRad51 was first modeled by aligning the core domains of SpRad51 onto those of the cryo-EM structure of the HsRAD51-dsDNA filament (PDB ID: 5H1C)[26] as a reference, using MODELLER. Then, the 1.5-fold-extended ssDNA was docked manually to the modeled SpRad51-dsDNA filament.

**Coimmunoprecipitation assay.** Rad51 (2.5 μM) and Swi5-Sfr1 (2.5 μM) were mixed in 100 μl of a buffer (30 mM Tris-HCl [pH 7.5], 20 mM KCl, 3 mM MgCl$_2$, 1 mM DTT, 5 % [v/v] glycerol and 0.05 % [w/v] NP-40) for 30 min at 4 °C. After taking 7.5 μl of the mixture from the reaction for an input sample, 5 μL of affinity-purified antibody against Rad51 from rabbit[29] was added and the mixtures were incubated for 1 h at 4 °C. Magnetic beads conjugated protein G (Dynabeads protein G, Thermo Fisher) was added and the reaction was incubated for another 1 h at 4 °C. Immunocomplexes were pulled down using a magnetic stand, eluted using SDS loading buffer and separated by SDS-PAGE. Proteins were detected by western blotting using the indicated antibodies, rat anti-Rad51 (1:6000), rabbit anti-Sfr1 (1:3000), and rabbit anti-Swi5 (1:500)[29]. The immunoblots were treated with anti-rabbit IgG-HRP (1:5000, GE Healthcare, NA934) or anti-rat IgG-HRP (1:5000, Jackson Laboratories, 712-035-153) using ImmunoStar Zeta (TAKARA), Proteins were detected by a LAS-4000mini scanner (Fuji). The asterisks indicate IgG heavy chain. All uncropped blots are included in the Source data file. Source data are provided as a Source data file.

**Statistics and reproducibility.** All data analyzed statistically are presented as averages of at least three independent experiments. Dot plots represent pooled data. Statistical tests were performed by two-tailed Student's $t$ test using Microsoft Excel software (Microsoft). All graphs were created using DataGraph software (Visual Data Tools, Inc). Reproducibility of EMSA (Fig. 6b and Supplementary Fig. 3a) and co-IP experiments (Supplementary Fig. 1d) was confirmed by at least two independent experiments.

**Reporting summary.** Further information on research design is available in the Nature Research Reporting Summary linked to this article.

## Data availability

The atomic coordinates (PDB ID:5H1B and 5H1C) deposited in the Protein Data Bank (www.wwpdb.org) were used for model constructions in Fig.1b and c, respectively. Raw data can be found in the Source data file. The Source data for Figs. 1d–f, 2a, b, 3b–f, 4b–d and 5a–d are provided as a Source data file_1. The Source data for Fig. S1a–d, Fig. S3a, b, Fig. 3b–f, Fig. S4 and Fig. 5a–d are provided as a Source data file_2. The Source data for

Table S7, Table S8, Table S9 and Table S10 are provided as a Source data file_3. The Source data for Figs. 1b and 7b (the structural model of Rad51-ssDNA filament) are provided as a Source data file_4. The Source data for Fig. 1c and 7b (the structural model of Rad51-dsDNA filament) are provided as a Source data file_5. The Source data for Fig. 7b (the structural model of C1 complex) is provided as a Source data file_6. The source data for Fig. 7b (the structural model of C2 complex) is provided as a Source data file_7. The Source data for Fig. S6a (the structural model of three-strand complex with a steric hindrance) is provided as a Source data file_8. The model structures have been submitted to the Biological Structure Model Archive (BSM-Arc) under BSM-ID BSM00017 (https://bsma.pdbj.org/entry/17). Thus, all data supporting the findings of this study are either available within the paper and its Supplementary Information files or canbe obtained from the authors upon reasonable request. Source data are provided with this paper.

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

## Acknowledgements

We thank all members of the Iwasaki Laboratory for discussion. This study was supported partly by Grants-in-Aids for Scientific Research on Innovative Areas (15H05974 to H.I.), for Scientific Research (A) (18H03985 to H.I.), for Scientific Research (B) (18H02371 to H.T. and 19H03160 to Y.M.), for Early-Career Scientists (19K16039 to K.I.), for Young Scientists (B) (17K15061 to B.A.) from the Japan Society for the Promotion of Science (JSPS).

## Author contributions

K.I., Y.M., M.T., and H.I. conceived the study and designed the research. K.I. performed all experiments. T. Mikawa helped the CD experiments. T. Mikawa, K.I., Y. Ku, S.K., T. Maki, B.A., H.T., M.T., and H.I. analyzed the data. Y. Ko. and M.I. constructed structural models. K.I., B.A., and H.I., wrote the paper.

## Competing interests

The authors declare no competing interests.
