## [Peer Review File · Nature Communications]

Reviewers' comments:

Reviewer #1 (Remarks to the Author):

RAD51-driven DNA strand exchange is a prerequisite for homology-directed DNA repair. However, the detailed mechanism and dynamics of this process remain an enigma. To address this question, Ito et al. used fluorescence-based real-time tracking methods to characterize three DNA-binding related mutant variants of Rad51 (L1 and L2 in site I, and S2 in site II). The same group has previously shown that Rad51-driven DNA strand exchange involves two distinct three-strand intermediates, namely C1 and C2. Here, the authors found that (1) L1 (R257) is important for the formation of C1 intermediate by facilitating dsDNA capture; (2) L2 (V295) promotes C1-C2 transition by stabilizing heteroduplex DNA in the C2 intermediate; and (3) S2 (R324 and K334) serves as an entry gate for both ssDNA and dsDNA during this process. Finally, a molecular simulation approach reveals the catalytic process of Rad51-mediated DNA strand exchange. The authors further propose that the insertion of the inter-triplet gap in the incoming duplex DNA by L2 Val-295 leads a stable elongated nucleoprotein filament for the homology search. The novelty of this manuscript is appreciated and will be of interest to others in the wider community. However, the following concerns need to be addressed to strengthen this manuscript.

Major concerns:

1. The authors propose that stabilization of the elongated dsDNA filament requires a deep insertion of L2 Val-295. It will be important to replace Val to Leu, instead of Ala, to verify this model.
2. The authors included the Swi5-Sfr1 auxiliary factor in various assays to see its effects on Rad51. It will be important to verify whether these Rad51 variants are defective in the interaction of Swi5-Sfr1.
3. Fig 1C-(b). Why is wild-type Rad51 ATPase activity not stimulated by ssDNA? It has been documented that the Rad51 ATPase activity can be stimulated by ssDNA (Sauvageau et. al., 2005, MCB). Consistent with this notion, why is S2 mutant defective in DNA binding but remains the same ATPase activity as wild-type counterpart (comparing Fig 1C-(b) and Fig 4A)?
4. Fig 4C & 5C. It will be important to include a negative control to verify whether this FRET-based assay could reflect the status of forming elongated presynaptic filaments by Rad51. One control can be done by setting up the reaction without the presence of ATP.
5. Fig 4C & 5C. To rule out the effect that Rad51 dynamic assembly influences the elongation between two FRET dyes, the authors should include AMP-PNP or Ca in the reaction to see whether they could obtain the same results.
6. Fig 5C-(d). How could we evaluate the difference between wild-type and L2 variant is meaningful? Appropriate controls or statistical analyses are required to make a strong argument.
7. A similar site II mutation has been examined in *S. cerevisiae* Rad51 (ScRad51). It has been documented that ScRad51 containing mutations in site II (II3A; R188, K361, and K371) retains the ability to form nucleoprotein filaments, but is fully defective in DNA strand exchange (Cloud et al., 2012, Science). Note that ScRad51 K361 and K371 correspond to K324 and K334 in SpRad51. The authors should compare and discuss their similarities and differences in the Discussion session.

Specific points:

1. Fig 3B-(c). This control experiment is not necessary to be included in the main figure. It should be combined with Fig. S1.

Reviewer #2 (Remarks to the Author):

Ito et al. analyze the role of two DNA binding sites in *Schizosaccharomyces pombe* Rad51 (site I and site II) in the catalysis of DNA strand exchange. The authors have used a variety of biochemical and biophysical approaches to examine the deficiencies in strand exchange activity of point mutants in the DNA binding motifs, including a mutant in each site I loop (L1 and L2) as well as a double mutant in site II (S2). Using fluorescence-based real-time assays, the authors show that the three mutants have distinct defects in processing the strand exchange intermediates (presynaptic filament, C1 (paranemic joint), and C2 (plectonemic joint)). 1) Rad51-S2 mutations abolish ss and dsDNA binding. Thus, the mutant is defective in both presynaptic filament formation and dsDNA capture, 2) Rad51-L1 mutant lacks dsDNA binding and is unable to form the C1 intermediate. 3) Rad51-L2 is able to produce C1, but unable to transition to C2, as verified by the accumulation of C1 product in an abortive strand exchange assay.

Taken together, the authors conclude that each of the examined components of the Rad51 binding sites have important but largely distinct roles in mediating Rad51-DNA interactions and transitions through the strand exchange process. This is a very nice study that complements what has been identified from the structural analyses of RecA and Rad51. It broadens our mechanistic understanding of DNA strand exchange carried out by general recombinases.

There are a few concerns and suggestions that need to be addressed.

Specific Comments

1. The title of the manuscript is a too broad to suggest that the paper reveals more about strand exchange than it actually does. Consider swapping "process" for "details" to more appropriately describe the work and try to incorporate the fact that the data focus on the DNA binding domains.
2. I recommend showing data for analysis of the three mutants (e.g., PAGE gel image, oligomerization status, and interaction with Swi5-Sfr1) somewhere in the manuscript.
3. Fig. 1C and page 6: If the authors want to claim that the ATPase of three mutants are comparable to WT Rad51 in the strand exchange assay condition, they should include ATP hydrolysis data in the presence of both ss and dsDNA. Also, the authors didn't discuss the lower ATPase activity of L2. In conjunction with the mutants, can the authors address if (C1 to C2) transition or (C2 to ssDNA release) may require ATP binding/hydrolysis?
4. Please explain in more detail how the amount of C1 % was calculated in Fig. 3A-f.
5. While the FRET data in Fig. 4C support Rad51 filament formation, it is not clear if the lower FRET signal is due to "elongation" of the DNA as the authors suggest. If the authors can provide more evidence (e.g., the decrease of the FRET signal matches what they would expect from 1.5x B-form DNA), please do so. Otherwise, consider revising. Also, it would be interesting if they would measure FRET change during other transition, such as (C2 to ssDNA-release).
6. Briefly explain why the S2 mutant should be a double-mutant.
7. It would be intriguing to examine if the mutants show any distinct defect in cells. Do the mutants (L1 and L2) have an ability to protect replication fork, that primarily requires Rad51 filament formation but not strand exchange activity? Discuss this issue at least.

Minor Comments

1. Page 12, paragraph 2; "not" should be "no."
2. Page 12, paragraph 3; "binging" should be "binding."
3. Page 19, the title of section that starts with "Analysis of association..." should lose the word "measuring."
4. Statistical significance to describe differences in activity would help support the author's claims. For example, the authors suggest the FRET signal is different between Rad51 and Rad51-L2 in Fig. 5C. Statistics would help here.
5. Fig 1 A (b) and (c); It is difficult to identify the locations of the residues in the structure. I would recommend simplifying the figure and highlight the residues.

Reviewer #3 (Remarks to the Author):

The manuscript by Ito et al uses a combination of real-time ensemble FRET and 2AP substitutions to reveal important findings regarding how Rad51 filaments promote DNA strand exchange. Although the presence of several three-strand intermediate states along the exchange pathway has already been demonstrated, this work provides the first unambiguous insight into how DNA binding sites I and II distinctively impact the formation of C1 and C2 intermediates. Using specific mutations, the work clarifies the role of L1 and L2 loops in Site I at different stages of the process and suggests a previously unknown role for Site II as an entry gate not only for dsDNA but also for ssDNA.

The manuscript is concise, well written and highlights the key findings. The work makes a key contribution to the field as it is likely that other RecA-like recombinases will function through a similar mechanism.

I support its publication after the authors have addressed the comments below:

1. Figure 2A-b and 2B-b show the time-dependent variation in fluorescein signal upon addition of WT and several mutants using the FRET pairing and displacement assay, respectively. Given that they have a FRET assay, I would like to have seen the simultaneous recording of both D and A, or alternatively, at least a control with no Rhodamine present. It is known that the addition of proteins might quench or enhance fluorescence and in the absence of these controls whether the changes in signal over time are due exclusively to FRET is not demonstrated.
2. It will have been useful if instead of relative fluorescence in Figure 2Ab and Figure 2Bb, the authors transformed this relative change into the amount of product formed. This will allow us to assess the DNA strand exchange efficiency monitored by these assays and compare it with literature values or with gel-based assays. As it is and without knowing the relative intensity of the D in the fully exchanged DNA strand it is not possible to know what is the level of final product formed.
3. Table S2 lists the values of rate constants obtained following the scheme in Figure 2C and these are plotted in panels 2Ca-f. The authors mentioned that these values have been obtained by 'simulating' each reaction (pairing and displacement) using the DynaFit software. How well the simulated curve fits the experimental curve is not shown, this should be graphically included in the supplementary section for each curve fitted using DynaFit.
4. Figure 2A-b shows a biphasic variation in the time-dependent signal for the WT and L2 mutants with and without S5S12. What is the mechanism behind this biphasic variation? The biphasic change does not seem to be present when following the pairing assays using 2AP.
5. In contrast to most EDTA-trapping experiments, Figure 2Ac shows at each injection point a much lower intensity than the threshold trace, what is causing this?

Minor points.

In most schematics, the D and A 'stars' are not easy to differentiate.

In figure 2Af, it is difficult to distinguish the colors of the different WT and variants.

Figure 5Cd please label the gel bands.

In Figure 4c and 5Cb-c the Y-axis for the spectra plots reads 'Normalized int.' It is unclear to me in which way it has been normalized, please clarify.

Thank you very much for handling our manuscript. We would also like to express our gratitude to the reviewers for their efforts in helping to improve our manuscript.

The issues raised are in black font and our responses are in blue font. All references to page numbers, line numbers and figures are accurate for the revised manuscript, not the initially submitted manuscript. Major changes have been highlighted in blue throughout the manuscript. Please note that the Discussion section has been re-structured extensively to include the reviewers' suggestions.

Reviewer #1 (Remarks to the Author):

RAD51-driven DNA strand exchange is a prerequisite for homology-directed DNA repair. However, the detailed mechanism and dynamics of this process remain an enigma. To address this question, Ito et al. used fluorescence-based real-time tracking methods to characterize three DNA-binding related mutant variants of Rad51 (L1 and L2 in site I, and S2 in site II). The same group has previously shown that Rad51-driven DNA strand exchange involves two distinct three-strand intermediates, namely C1 and C2. Here, the authors found that (1) L1 (R257) is important for the formation of C1 intermediate by facilitating dsDNA capture; (2) L2 (V295) promotes C1-C2 transition by stabilizing heteroduplex DNA in the C2 intermediate; and (3) S2 (R324 and K334) serves as an entry gate for both ssDNA and dsDNA during this process. Finally, a molecular simulation approach reveals the catalytic process of Rad51-mediated DNA strand exchange. The authors further propose that the insertion of the inter-triplet gap in the incoming duplex DNA by L2 Val-295 leads a stable elongated nucleoprotein filament for the homology search. The novelty of this manuscript is appreciated and will be of interest to others in the wider community. However, the following concerns need to be addressed to strengthen this manuscript.

Major concerns:

1. The authors propose that stabilization of the elongated dsDNA filament requires a deep insertion of L2 Val-295. It will be important to replace Val to Leu, instead of Ala, to verify this model.

Although we could not directly measure the stability of elongated dsDNA filaments, our data showed that the valine-to-alanine substitution impaired the elongation of dsDNA by Rad51. This was explained with the following rational.

Both RecA-dsDNA crystallography data (Chen et al. *Nature* 2008: **453**, 489-4) and cryo-EM observations of human Rad51-dsDNA complexes (Xu et al. *NSMB* 2017: **24**, 40-46) demonstrated that the corresponding valine side-chain inserts in-between two triplets to stabilize dsDNA. These results indicated that dsDNA elongation by the valine insertion is closely related to stabilization of the elongated dsDNA filament. The valine-to-alanine substitution we employed reduces the size and length of the side-chain and thus results in a shallow insertion into dsDNA, which is naturally expected to decrease the level of the separation between two triplets. This is our interpretation.

In contrast to the valine-to-alanine mutation, the valine-to-leucine substitution suggested by this Reviewer lengthens the side chain and increases its size. Therefore, this mutation is not expected to result in reduced elongation of dsDNA by the mutant Rad51. Furthermore, this substitution will not necessarily enhance dsDNA elongation since the valine side-chain of the wild-type protein is considered to already be optimized for dsDNA elongation, as evidenced by the high degree of conservation among Rad51 proteins.

Therefore, we do not think that the proposed experiment will provide a conclusive answer as to whether stabilization of the elongated dsDNA filament requires a deep insertion of Val-295. However, because we do not have an established methodology to directly measure the stability of elongated dsDNA filaments, we accept that this Reviewer's point is valid. As a result, we have revised the corresponding part in the Discussion section to tone-down our conclusion (**Page 15 Lines 27-34 and Page 18 Lines 5-15**).

2. The authors included the Swi5-Sfr1 auxiliary factor in various assays to see its effects on Rad51. It will be important to verify whether these Rad51 variants are defective in the interaction of Swi5-Sfr1.

We examined the physical interaction between Rad51 variants and Swi5-Sfr1 by a colP experiment. We found no defect in the physical interaction. This result has been described in the Results section (**Page 6 Lines 20-22**) and is shown in Supplementary Fig. 1d.

3. Fig 1C-(b). Why is wild-type Rad51 ATPase activity not stimulated by ssDNA? It has been documented that the Rad51 ATPase activity can be stimulated by ssDNA (Sauvageau et. al., 2005, MCB). Consistent with this notion, why is S2 mutant defective in DNA binding but remains the same ATPase activity as wild-type counterpart (comparing Fig 1C-(b) and Fig 4A)?

[REDACTED]

Please see the attached figure (Fig. 4 of Sauvageau et. al., 2005, MCB). Panel A shows that the ATPase activity of *S. pombe* Rad51 in the absence of DNA is considerably high, with only a slight stimulation observed by the inclusion of ssDNA. This is in complete agreement with our data: the k_{cat} of Rad51 alone is 0.163 min^{-1} , and this increases to 0.178 min^{-1} in the presence of ssDNA. This is in contrast to *S. cerevisiae* and human Rad51, whose ATPase activity is strongly stimulated (several fold) by ssDNA (Sung et al. 1994, Science; Baumann et al. 1996, Cell).

As for the ATPase activity of the S2 mutant. As mentioned above, *S. pombe* Rad51 already displays a high level of ATPase activity even in the absence of ssDNA. Therefore, it is not surprising that the S2 mutant retains a high level of ATPase activity despite having reduced DNA binding activity. Related to this point, the S2 mutant shows reduced ss- and dsDNA binding, but does not completely lose it (Fig. 5a, b, 6a, b, Supplementary Fig.

2). Thus, it is important to note that, in an effort to separate any potential defect in ATP hydrolysis from the DNA binding defect, we employed an experimental condition where excess amounts of ss- or dsDNA were included to the reaction to promote the binding of DNA by the S2 mutant.

4. Fig 4C & 5C. It will be important to include a negative control to verify whether this FRET-based assay could reflect the status of forming elongated presynaptic filaments by Rad51. One control can be done by setting up the reaction without the presence of ATP.

This is a great suggestion and we have now conducted these experiments. The FRET efficiencies of three ssDNA-only reactions (no nucleotide, ATP or AMP-PNP) were the same. The FRET efficiency of a reaction containing ssDNA+Rad51 without nucleotide was reduced compared to the ssDNA-only reaction. However, this reduction was much less than

that observed in the reaction containing ssDNA+Rad51+ATP, which is consistent with the notion that the observed change in FRET reflects filament elongation. The same is true for the dsDNA reaction. We described these results in the Results section (**Page 13. Lines 10-11, Page 14 Line 14-16**) and have shown them in Supplementary Fig. 4a, b for ssDNA and Supplementary Fig.4c, d for dsDNA.

5. Fig 4C & 5C. To rule out the effect that Rad51 dynamic assembly influences the elongation between two FRET dyes, the authors should include AMP-PNP or Ca in the reaction to see whether they could obtain the same results.

Thank you for this suggestion. A similar trend was observed in this assay when we included AMP-PNP instead of ATP. This result has been described in the Results section (**Page 13. Lines 11-19, Page 14 Lines 20-23**) and is shown in Supplementary Fig. 4.

6. Fig 5C-(d). How could we evaluate the difference between wild-type and L2 variant is meaningful? Appropriate controls or statistical analyses are required to make a strong argument.

We have shown the P values obtained by T-test. The P values of all combinations were extremely low ($P < 0.0002$), indicating that the difference between wild-type and L2 is statistically significant.

7. A similar site II mutation has been examined in *S. cerevisiae* Rad51 (ScRad51). It has been documented that ScRad51 containing mutations in site II (II3A; R188, K361, and K371) retains the ability to form nucleoprotein filaments, but is fully defective in DNA strand exchange (Cloud et al., 2012, Science). Note that ScRad51 K361 and K371 correspond to K324 and K334 in SpRad51. The authors should compare and discuss their similarities and differences in the Discussion session.

[REDACTED]

Thank you for this suggestion. As pointed out by this Reviewer, ScRad51-K361 and -K371 correspond to K324 and K334 in SpRad51. Although Cloud et al. (2012) seems to argue that II3A mutant has no severe defect in ssDNA binding, they actually reported that the II3A mutant shows a reduced binding affinity for both ssDNA and dsDNA (~2.5-fold higher K_D according to Fig.1B in their paper, please see the attached figure) as judged by EMSA. Importantly, they show that the presynaptic filament formed by II3A cannot bind to heterologous dsDNA. These

characterizations of II3A are essentially consistent with our presented data. However, in our study, we extended the analysis further to characterize the step that the Site II mutation affects in the three-step model of strand exchange. Note that at the time when Cloud et al. (2012) published their study, the three-step model had not been proposed yet. We have incorporated this information into the Discussion section (**Page 17. Lines 27-34**).

Specific points:

1. Fig 3B-(c). This control experiment is not necessary to be included in the main figure. It should be combined with Fig. S1.

While we appreciate this comment, we feel that Fig. 3B-(c) serves a greater purpose than just a control experiment. It is important for readers to understand the actual experiment because they may otherwise misinterpret the results. The critical difference is that an increase in FRET is observed in the standard pairing assay when the C1 intermediate is formed, whereas an increase in quenching in the 2AP assay is only observed when the C2 intermediate forms. This difference is also reflected in the different shapes of the corresponding curves. To emphasize this difference, we would like to keep Fig. 4b as part of the main figure.

Reviewer #2 (Remarks to the Author):

Ito et al. analyze the role of two DNA binding sites in *Schizosaccharomyces pombe* Rad51 (site I and site II) in the catalysis of DNA strand exchange. The authors have used a variety of biochemical and biophysical approaches to examine the deficiencies in strand exchange activity of point mutants in the DNA binding motifs, including a mutant in each site I loop (L1 and L2) as well as a double mutant in site II (S2). Using fluorescence-based real-time assays, the authors show that the three mutants have distinct defects in processing the strand exchange intermediates (presynaptic filament, C1 (paranemic joint), and C2 (plectonemic joint)). 1) Rad51-S2 mutations abolish ss and dsDNA binding. Thus, the mutant is defective in both presynaptic filament formation and dsDNA capture, 2) Rad51-L1 mutant lacks dsDNA binding and is unable to form the C1 intermediate. 3) Rad51-L2 is able to produce C1, but unable to transition to C2, as verified by the accumulation of C1 product in an abortive strand exchange assay.

Taken together, the authors conclude that each of the examined components of the Rad51 binding sites have important but largely distinct roles in mediating Rad51-DNA interactions and transitions through the strand exchange process. This is a very nice study that complements what has been identified from the structural analyses of RecA and Rad51. It broadens our mechanistic understanding of DNA strand exchange carried out by general recombinases.

There are a few concerns and suggestions that need to be addressed.

Specific Comments

1. The title of the manuscript is a too broad to suggest that the paper reveals more about strand exchange than it actually does. Consider swapping “process” for “details” to more appropriately describe the work and try to incorporate the fact that the data focus on the DNA binding domains.

Thank you for this suggestion. We have revised the title as follows:
“Real-time tracking reveals catalytic roles for the two DNA binding sites of Rad51”

2. I recommend showing data for analysis of the three mutants (e.g., PAGE gel image, oligomerization status, and interaction with Swi5-Sfr1) somewhere in the manuscript.

Thank you for this suggestion. We have incorporated an SDS-PAGE image of the purified proteins, as well as CD spectra at 25 °C and thermal stability. The results of the latter experiments suggest that the three mutants have no significant changes in their overall structure. In addition, we performed colP experiments to test the interactions with Swi5-Sfr1 and the result indicated that the three Rad51 mutants are proficient in the Swi5-Sfr1 interaction. These results have now been incorporated as Supplementary Fig. 1a-d and been described in the first part of Results section (Page 6 Lines 16-22).

3. Fig. 1C and page 6: If the authors want to claim that the ATPase of three mutants are comparable to WT Rad51 in the strand exchange assay condition, they should include ATP hydrolysis data in the presence of both ss and dsDNA. Also, the authors didn't discuss the lower ATPase activity of L2. In conjunction with the mutants, can the authors address if (C1 to C2) transition or (C2 to ssDNA release) may require ATP binding/hydrolysis?

3-1) If the authors want to claim that the ATPase of three mutants are comparable to WT Rad51 in the strand exchange assay condition, they should include ATP hydrolysis data in the presence of both ss and dsDNA.

The ATPase assay was conducted in the presence of both ss- and dsDNA. Briefly, only the L1 mutant showed lower ATPase activity in this condition. However, the ATPase activity of the L1 mutant was restored to WT levels in the presence of Swi5-Sfr1, consistent with our assertion that the strand exchange defect observed in Fig. 1d is not due to impaired ATP hydrolysis. We have shown the result in Fig 1f and Supplementary Table 1, and a detailed description of these findings can now be found in the Results section (**Page 7 Lines 6-14**).

3-2) Also, the authors didn't discuss the lower ATPase activity of L2. In conjunction with the mutants, can the authors address if (C1 to C2) transition or (C2 to ssDNA release) may require ATP binding/hydrolysis?

Both the C1-to-C2 transition and C2-to-ssDNA release require ATP hydrolysis, as was previously shown (Ito et al 2018). We agree with this Reviewer that these steps of strand exchange are likely related to ATP hydrolysis. These points have now been mentioned in the Discussion (**Page 18 Lines 15-32**).

4. Please explain in more detail how the amount of C1 % was calculated in Fig. 3A-f. "Amount of C1 %" was calculated by the same method as the pairing assay, using the equation below, where S_{before} is the amount of substrate before addition of EDTA and S_{after} is the amount of substrate 200 seconds after addition of EDTA:
Amount of C1 % = $100 \times (S_{\text{after}} - S_{\text{before}}) / (100 - S_{\text{before}})$

We have now included this information in the Methods section (**Page 22 Lines 2-6**).

5. While the FRET data in Fig. 4C support Rad51 filament formation, it is not clear if the lower FRET signal is due to "elongation" of the DNA as the authors suggest. If the authors can provide more evidence (e.g., the decrease of the FRET signal matches what they would expect from 1.5x B-form DNA), please do so. Otherwise, consider revising. Also, it would be interesting if they would measure FRET change during other transition, such as (C2 to ssDNA-release).

We calculated the relative lengths of DNA extended by Rad51 binding in Fig. 5d. The results show that the lengths of extended ssDNA in the presence of Swi5-Sfr1 correspond to 1.6x B-form DNA. In addition, when we performed the same experiment with AMP-PNP (as suggested by Reviewer 3), a similar extension was observed. Furthermore, we calculated the elongation for dsDNA shown in Fig. 6c. Wild-type Rad51 elongated dsDNA to ~1.4x B-form DNA in the presence of Swi5-Sfr1 or with AMP-PNP instead of ATP. The values of elongation relative to B-form DNA obtained by this assay agree with previously reported values obtained by structural analysis (e.g. Chen et al 2008 Nature). This strongly suggests that the FRET signal is due to DNA elongation caused by the binding of Rad51.

During this analysis, we discovered a miscalculation. Having corrected this mistake, we found that the L2 mutant has a slight defect in ssDNA elongation in addition to the defect in dsDNA elongation that we originally reported. These results are shown in Supplementary Fig. 4 and have been described in the corresponding parts of the Results section (**Page 13 Lines 10-19, Page 14 Lines 12-23**). We have also described the calculation procedures in the Methods section (**Page 25 Lines 7-21**).

While we agree that it would be very interesting, it is impossible to measure FRET changes during the transition of C2-to-ssDNA release because one cannot pinpoint the origin of the signal change. Indeed, we did previously try this but were unable to determine whether the changes were due to Rad51 binding per se or by real FRET.

6. Briefly explain why the S2 mutant should be a double-mutant.

These two basic amino acid residues have been proposed to play an equally important role in DNA binding by Kurumizaka et al. (1999, Arch Biochem Biophys) and by Chen et al. (2008, Nature). We therefore simultaneously substituted both sites to Ala in the S2 mutant. This point has now been described in the Results (**Page 6 Lines 13-15**) and Discussion (**Page 15 Lane 9**) sections.

7. It would be intriguing to examine if the mutants show any distinct defect in cells. Do the mutants (L1 and L2) have an ability to protect replication fork, that primarily requires Rad51 filament formation but not strand exchange activity? Discuss this issue at least.

We agree that it would be interesting to examine whether our mutants have the ability to protect replication forks. However, we do not have a specific assay system to examine the ability of Rad51 to protect replication forks. Therefore, we unfortunately cannot conduct such an experiment at this moment in time. However, we have discussed this possibility in the Discussion section as suggested by this Reviewer (**Page 18 Line 33- Page 19 Line 2**).

Minor Comments

1. Page 12, paragraph 2; “not” should be “no.”

Thank you for bringing this typo to our attention. Corrected.

2. Page 12, paragraph 3; “binging” should be “binding.”

Thank you for bringing this typo to our attention. Corrected.

3. Page 19, the title of section that starts with “Analysis of association...” should lose the word “measuring.”

Thank you, this has been corrected.

4. Statistical significance to describe differences in activity would help support the author’s claims. For example, the authors suggest the FRET signal is different between Rad51 and Rad51-L2 in Fig. 5C. Statistics would help here.

We performed a T-test and the result was incorporated in the revised version.

5. Fig 1 A (b) and (c); It is difficult to identify the locations of the residues in the structure. I would recommend simplifying the figure and highlight the residues.

Thank you for this suggestion. These figures have been modified. We hope it is now easier to identify the residues of interest.

Reviewer #3 (Remarks to the Author):

The manuscript by Ito et al uses a combination of real-time ensemble FRET and 2AP substitutions to reveal important findings regarding how Rad51 filaments promote DNA strand exchange. Although the presence of several three-strand intermediate states along the exchange pathway has already been demonstrated, this work provides the first unambiguous insight into how DNA binding sites I and II distinctively impact the formation of C1 and C2 intermediates. Using specific mutations, the work clarifies the role of L1 and L2 loops in Site I at different stages of the process and suggests a previously unknown role for Site II as an entry gate not only for dsDNA but also for ssDNA.

The manuscript is concise, well written and highlights the key findings. The work makes a key contribution to the field as it is likely that other RecA-like recombinases will function through a similar mechanism.

I support its publication after the authors have addressed the comments below:

1. Figure 2A-b and 2B-b show the time-dependent variation in fluorescein signal upon addition of WT and several mutants using the FRET pairing and displacement assay, respectively. Given that they have a FRET assay, I would like to have seen the simultaneous recording of both D and A, or alternatively, at least a control with no Rhodamine present. It is known that the addition of proteins might quench or enhance fluorescence and in the absence of these controls whether the changes in signal over time are due exclusively to FRET is not demonstrated.

We previously demonstrated that *S. pombe* Rad51 does not affect fluorescence emission of FAM or quenching efficiency of fluorescein by rhodamine in both assays (Supplementary Fig 5 and Supplementary Table 6 in Ito et al 2018, NSMB). This has now been described in the main text (**Page 7 Lines 34-35**) and *Methods* (**Page 21 Lines 9-30**).

2. It will have been useful if instead of relative fluorescence in Figure 2Ab and Figure 2Bb, the authors transformed this relative change into the amount of product formed. This will allow us to assess the DNA strand exchange efficiency monitored by these assays and compare it with literature values or with gel-based assays. As it is and without knowing the relative intensity of the D in the fully exchanged DNA strand it is not possible to know what is the level of final product formed.

Thank you for this suggestion. We have changed relative fluorescence in Figure 2a and Figure 2b to "Substrate %" and "Product %," respectively. We have also described the calculation procedures in the *Methods* section (**Page 21 Lines 9-30**).

3. Table S2 lists the values of rate constants obtained followed the scheme in Figure 2C and these are plotted in panels 2Ca-f. The authors mentioned that these values have been obtained by 'simulating' each reaction (pairing and displacement) using the DynaFit software. How well the simulated curve fits the experimental curve is not shown, this should be graphically included in the supplementary section for each curve fitted using DynaFit.

We apologize for this oversight. The residuals between experimental data of the DNA strand pairing assay in Figure 2a and a theoretical curve obtained by simulation using DynaFit (which corresponds to Figure 2c), along with determination coefficients, are now shown in Supplementary Fig. 2. We have also described this point in the corresponding part of the *Results* section (**Page 8 Lines 7-9, Page 9 Lines 3-4, and Page 9 Lines 29-31**).

4. Figure 2A-b shows a biphasic variation in the time-dependent signal for the WT and L2 mutants with and without S5S12. What is the mechanism behind this biphasic variation? The biphasic change does not seem to be present when following the pairing assays using 2AP.

The FRET-based pairing assay (Fig. 2a) encompasses multiple later steps of the strand exchange reaction, namely progression of the C1-to-C2 transition and the formation of final products. Due to the processes involved, the curve of wild-type Rad51 shows multiple phases (because it is a 3-step reaction, the curve is actually triphasic, not biphasic). In the Rad51-L2 reaction, the C1-to-C2 transition is significantly slower than that of wild type, so a large difference is seen between the two reactions. Wild-type Rad51 has a reaction curve that gradually declines over the course of the reaction. This contrasts with Rad51-L2, which resembles wild type at the beginning of the reaction but then shows a reduced slope, indicating that the reaction does not progress further. Swi5-Sfr1 strongly promotes the C1-to-C2 transition of both wild type and the L2 mutant, as well as the formation of end products in the case of wild-type Rad51. Therefore, Swi5-Sfr1 changes the shapes of the curve of the reactions, as we reportedly previously (Ito et al 2018 NSMB).

In experiments involving 2AP, quenching occurs only when a heteroduplex is formed (please see the schematic in Fig. 4a), which is different from the FRET assay. The formation of the C1 intermediate is therefore not detected by the 2AP assay. Consequently, the 2AP reaction shows a quasi-linear curve.

We have added this information into the Results section (**Page 8 Lines 12-14, Page 8 Lines 33-35 and Page 11 Lines 9-11**).

5. In contrast to most EDTA-trapping experiments, Figure 2Ac shows at each injection point a much lower intensity than the threshold trace, what is causing this?

This apparent difference is due to a difference in data presentation; the Y-axis in Fig 2 was relative fluorescence change and that in Fig 3 was normalized intensity. We have unified them to "substrate/product %" for Y-axis in the whole manuscript in order to compare results more easily. Accordingly, we have added the formula to calculate them in *Methods* (**Page 21 Lines 9-30**).

Minor points.

In most schematics, the D and A 'stars' are not easy to differentiate. Thank you for pointing this out. We have enlarged these symbols.

In figure 2Af, it is difficult to distinguish the colors of the different WT and variants. We have modified Fig. 2 to better differentiate the colors.

Figure 5Cd please label the gel bands.

Thank you. We have annotated the gel picture to specify free dsDNA and Rad51+dsDNA complexes in the gel.

In Figure 4c and 5Cb-c the Y-axis for the spectra plots reads 'Normalized int.' It is unclear to me in which way it has been normalized, please clarify.

'Normalized Int.' was calculated using the equation below:

(Normalized Int.) = (fluorescence intensity at x nm of each condition)/(fluorescence intensity at 525 nm of DNA only)

This information, as well as other important calculation, have been described in the Methods section (**Page 24 Lines 5-7**).

REVIEWERS' COMMENTS:

Reviewer #1 (Remarks to the Author):

The revised manuscript has addressed all the concerns raised by this reviewer. Only a few typos as below:

1. Page 15, line 29; "Swi5-Sr1" should be "Swi5-Sfr1".
2. The figure legend of supplementary Fig 4; "Supplementary Fig. 4b" should be "Supplementary Fig. 4"
3. Figure 1a; the shadow of Arg-257 (L1) and Arg-324/Lys-334 (Site II) is misaligned.

Reviewer #2 (Remarks to the Author):

Authors have done a fine job addressing the previous critique.

The study adds significantly to our understanding of recombinase mechanisms and publication in Nature Communications is fully warranted.

Reviewer #3 (Remarks to the Author):

The authors have addressed all my comments and when appropriate, they have modified existing figures and added additional information that has improved the manuscript. I have no hesitation to recommend the manuscript for publication in its current format

Responses to the Reviewer's comments.

Reviewer #1 (Remarks to the Author):

The revised manuscript has addressed all the concerns raised by this reviewer. Only a few typos as below:

1. Page 15, line 29; "Swi5-Sr1" should be "Swi5-Sfr1".
Thank you. We have corrected it.
2. The figure legend of supplementary Fig 4; "Supplementary Fig. 4b" should be "Supplementary Fig. 4"
Thank you. We have corrected it.
3. Figure 1a; the shadow of Arg-257 (L1) and Arg-324/Lys-334 (Site II) is misaligned.
Thank you. We revised the figure.